# Thermally-assisted photosensitized emission in a trivalent terbium complex

Yuichi Kitagawa [1,2✉], Kaori Shima[3], Takuma Nakai[3], Marina Kumagai[3], Shun Omagari[4],
Pedro Paulo Ferreira da Rosa[3], Sunao Shoji [1,2,5], Koji Fushimi [1,2] & Yasuchika Hasegawa [1,2✉]

Luminescent lanthanide complexes containing effective photosensitizers are promising materials for use in displays and sensors. The photosensitizer design strategy has been studied for developing the lanthanide-based luminophores. Herein, we demonstrate a photosensitizer design using dinuclear luminescent lanthanide complex, which exhibits thermally-assisted photosensitized emission. The lanthanide complex comprised Tb(III) ions, six tetramethylheptanedionates, and phosphine oxide bridge containing a phenanthrene frameworks. The phenanthrene ligand and Tb(III) ions are the energy donor (photosensitizer) and acceptor (emission center) parts, respectively. The energy-donating level of the ligand (lowest excited triplet ($T_1$) level = 19,850 cm$^{-1}$) is lower than the emitting level of the Tb(III) ion ($^5D_4$ level = 20,500 cm$^{-1}$). The long-lived $T_1$ state of the energy-donating ligands promoted an efficient thermally-assisted photosensitized emission of the Tb(III) acceptor ($^5D_4$ level), resulting in a pure-green colored emission with a high photosensitized emission quantum yield (73%).

[1] Faculty of Engineering, Hokkaido University, N13W8, Kita-ku, Sapporo, Hokkaido 060–8628, Japan. [2] Institute for Chemical Reaction Design and Discovery (WPI-ICReDD), Hokkaido University, N21W10, Sapporo, Hokkaido 001-0021, Japan. [3] Graduate School of Chemical Sciences and Engineering, Hokkaido University, N13W8, Sapporo, Hokkaido 060-8628, Japan. [4] Department of Materials Science and Engineering, Tokyo Institute of Technology, Ookayama 2-12-1-S8-44, Meguro-ku, Tokyo 152-8552, Japan. [5] Department of Engineering, Nara Women's University, Kitauoya Nishimachi, Nara 630-8506, Japan. ✉email: y-kitagawa@eng.hokudai.ac.jp; hasegaway@eng.hokudai.ac.jp

Highly luminescent molecules have become increasingly important for the development of display and sensing devices[1–6]. Numerous studies on luminescent molecular materials based on organic compounds and metal complexes have been reported[7–10]. Among these materials, visible luminescent lanthanide (where Ln(III) = Tb(III) and Eu(III)) complexes are considered promising candidates for highly luminescent molecules with high color purity originating from the intra-4f-orbital transitions[11–13]. However, they exhibit a small absorption coefficient ($\varepsilon = 0.1$–$10\,M^{-1}\,cm^{-1}$), which is mitigated by photosensitized energy transfer from organic ligands with a larger absorption coefficient ($\varepsilon = 10^3$–$10^5\,M^{-1}\,cm^{-1}$). Therefore, effective photosensitizer design is crucial for realizing strong lanthanide emissions.

The organic ligands undergo intersystem crossing (ISC) from the lowest singlet excited state ($S_1$) to the lowest triplet excited state ($T_1$) after excitation, thereby transferring their electronic energy to the Ln(III) ion. Latva et al. conducted a detailed investigation of the relationship between the photosensitized emission efficiency and $T_1$ level using green luminescent Tb(III) and amino-carboxylate-typed ligands[14]. They suggested that the energy of the $T_1$ level should be enough higher than that of the emitting level of Tb(III) ions ($^5D_4$: $20,500\,cm^{-1}$, Supplementary Note 1 and Fig. S1) for strong Ln(III) emission (Fig. 1a; required energy gap between donor and acceptor in case of Tb(III) complexes >1850 cm$^{-1}$ (Latva's empirical rule)). This photosensitized energy transfer system, requiring a high $T_1$ level, causes a strong restriction of the organic ligand designs in lanthanide complexes[15–18].

Herein, we focused on the long-lived excited organic donor system to break this photosensitizer design rule of luminescent Ln(III) complexes. Theoretical calculations have suggested that the $T_1$-Ln(III) energy transfer rate is much higher than the inverse of the lifetime of the excited states of Ln(III) ions[19]. The long $T_1$ lifetime should allow the efficient use of Ln(III) emitting photons, even in the case of a low $T_1$ level, when an excited equilibration between $T_1$ and Ln(III) emitting states is formed[20–26]. In this study, we demonstrated the photosensitized emission of the Tb(III) complex with a $T_1$ level of an organic ligand lower than the Tb(III) emitting level for the first time, using the long-lived excited organic ligands (Fig. 1b and Supplementary Note 2).

To demonstrate our conceptual strategy, we designed the seven-coordinated Tb(III) complexes with a 2,2,6,6-tetramethyl-3,5-heptanedionate (tmh) and bidentate phosphine oxide-containing phenanthrene framework (dpph, Fig. 1c). A density functional theory calculation indicated that the $T_1$ level of dpph is lower than that of a Tb(III) ion. The phosphine oxide-containing polyaromatic hydrocarbon framework also provides long-lived localized $T_1$ states in lanthanide complexes, which function as effective energy donors[20,27,28]. Two-sided tmh ligands encapsulate the dpph ligand, thereby extending dpph's $T_1$ lifetime[29]. The Lu(III) complex with a closed 4f-electronic configuration was prepared to estimate dpph's energy level and excited lifetime in an Ln(III) complex (Fig. 1d)[30]. The photosensitization mechanism presents new frontiers in the fields of molecular lanthanide photophysics and photofunctional material science.

## Results and discussion

**Coordination structure.** The Tb(III)–Tb(III) and Lu(III)–Lu(III) dinuclear complexes were prepared by the complexation of [Tb$_2$(tmh)$_6$] and [Lu(tmh)$_3$] with dpph in methanol, respectively (where [Tb$_2$(tmh)$_6$(dpph)]: **Tb-dpph**, and [Lu$_2$(tmh)$_6$(dpph)]: **Lu-dpph**). Single crystals of the dinuclear Tb(III) complex were obtained by recrystallization from the methanol solution. The crystal structure of **Tb-dpph**, shown in Fig. 2, was found to be triclinic, with the space group being P-1 (for the crystallographic data, see Table S1, ESI†). The coordination site in the Tb(III) complex comprised three tmh ligands and one phosphine oxide ligand. The single-crystal structure of the as-obtained **Lu-dpph** is almost the same as that of **Tb-dpph** (Fig. S2).

**Photophysical properties of ligand-excited states.** The emission spectrum of **Lu-dpph**, in degassed condition, is shown in Fig. 3a (solid line). The **Lu-dpph** shows a broad band at around 550 nm, which originated from π-π* transition of dpph ligand moiety (Supplementary Note 3 and Fig. S3–S5). The emission spectrum was deconvoluted into three vibronic bands using the software (OriginPro 2021b), the spectrum in wavenumber scale, and by fitting the peak profile using Gaussian functions (Fig. 3a, broken line). The deconvolution results in the three vibronic bands were designated as 0–0 ($19,850\,cm^{-1}$), 0–1 ($18,670\,cm^{-1}$), and 0–2 ($17,390\,cm^{-1}$). Thus, the $T_1$ level of the dpph ligand in **Lu-dpph**

### (a) General lanthanide chemistry

### (b) This study

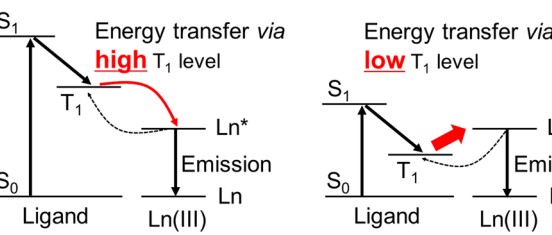

### (c) Tb(III) complex for proof of concept

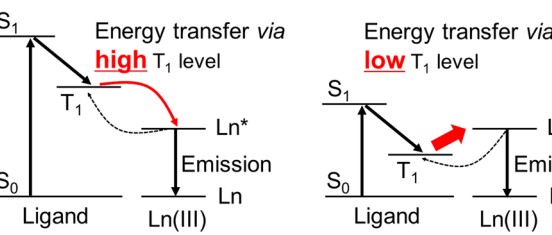

### (d) Lu(III) complex for estimating ligand excited state

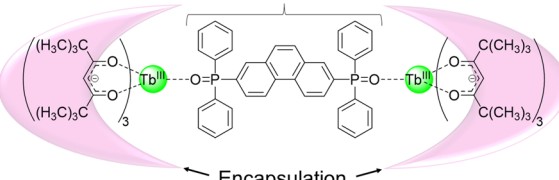

**Fig. 1 Photosensitizer design strategies.** Schematic photosensitized emission mechanism based on energy transfer from high $T_1$ level (**a**, general lanthanide chemistry) and low $T_1$ level (**b**, this study). The chemical structures of Tb(III) complex for proof of concept (**c**) and Lu(III) complex for estimating ligand-excited states (**d**).

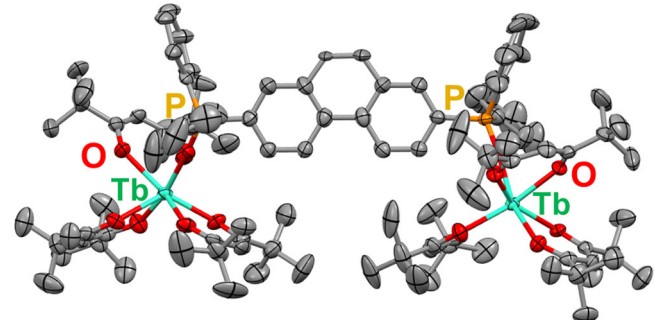

**Fig. 2 Crystal structure.** ORTEP drawings (ellipsoids set at 50% probability) of **Tb-dpph** without hydrogen atoms. Gray spheres represent carbon; red spheres, oxygen; orange spheres, phosphorus; light green spheres, terbium.

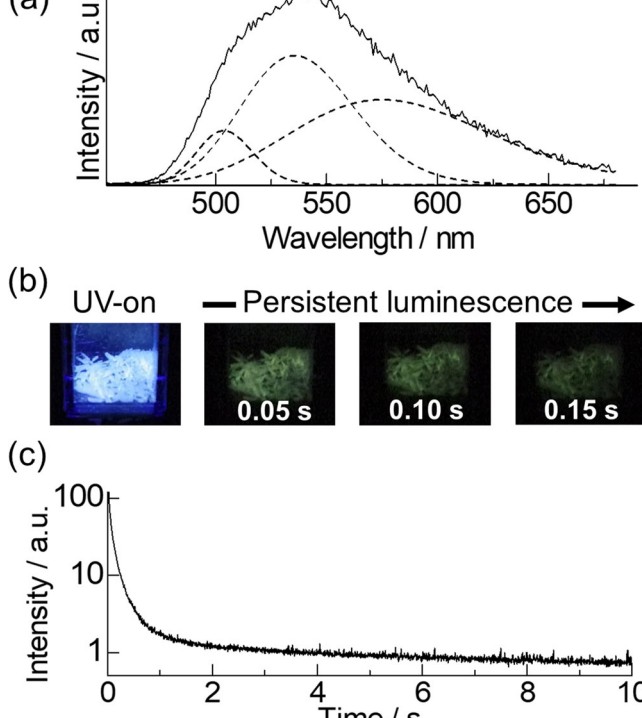

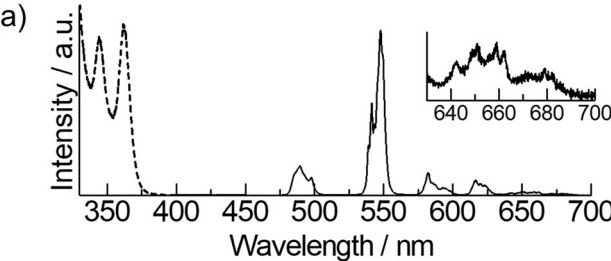

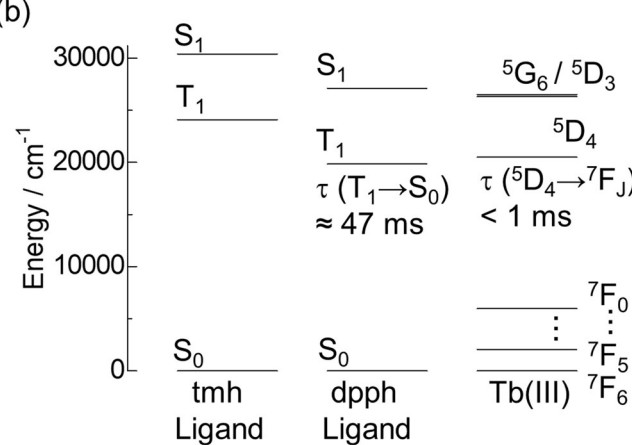

**Fig. 3 Photophysical properties of a ligand localized excited state.** The emission spectrum of **Lu-dpph** (**a**: $\lambda_{ex}$ = 400 nm; delay: 80 ms; 100 K). Emission images of **Lu-dpph** excited using UV-light (**b**: $\lambda_{ex}$ = 375 nm; 293 K) under vacuum conditions. Emission-decay curves of **Lu-dpph** (**c**: $\lambda_{ex}$ = 400 nm; $\lambda_{em}$ = 530 nm; 293 K).

**Fig. 4 Photophysical properties of a trivalent terbium complex.** Emission (solid line) and excitation (broken line) spectra of **Tb-dpph** (**a**: $\lambda_{ex}$ = 356 nm; $\lambda_{em}$ = 548 nm; 293 K). **b** The energy diagram for **Tb-dpph**.

was determined to be 19,850 cm$^{-1}$ using band-deconvolution analysis. The emission photograph of **Lu-dpph** is shown in Fig. 3b, where it shows a green persistent luminescence.

The emission durability of **Lu-dpph** was evaluated using time-resolved emission spectroscopy (Fig. 3c), yielding characteristic emission-decay curves for persistent-emission materials (Supplementary Note 4 and Figs. S6–S8). Herein, the emission lifetimes were estimated using triple exponential functions ($\tau_1$ = 16 ms (70 %), $\tau_2$ = 83 ms (27%), and $\tau_3$ = 450 ms (3%)). The average $\pi$–$\pi^*$ emission lifetime of the dpph ligand in **Lu-dpph** was estimated to be 47 ms, which is characteristic among the $T_1$ lifetime of organic ligands in lanthanide complexes at room temperature[30–33]. The long $T_1$ lifetime in the dpph moiety was ascribed to the rigid isolated polyaromatic structure encapsulated in the tmh ligands, which suppressed the non-radiative deactivation pathways[29,34,35]. These results indicate the construction of an energy-donating system with a long $T_1$ lifetime in **Tb-dpph**. This dpph $T_1$ lifetime (47 ms) is significantly longer than the 4f–4f emission lifetimes of reported Tb(III) complexes[11,36].

**Photophysical properties of a trivalent terbium complex.** The emission and excitation spectra of **Tb-dpph** in degassed conditions are shown in Fig. 4a. Sharp emission bands at 490, 548, 583, 616, 651, and 679 nm were observed for **Tb-dpph**, which are assigned to the $^5D_4 \rightarrow {}^7F_6$, $^5D_4 \rightarrow {}^7F_5$, $^5D_4 \rightarrow {}^7F_4$, $^5D_4 \rightarrow {}^7F_3$, $^5D_4 \rightarrow {}^7F_2$, and $^5D_4 \rightarrow {}^7F_{1,0}$ transitions of Tb(III), respectively. The observed excitation spectral bands at 344 and 362 nm are consistent with the absorption bands of the dpph ligand (Fig. S9), indicating energy transfer from the $\pi$-conjugated dpph ligand to Tb(III). The emission quantum yield and emission lifetime of **Tb-dpph** excited by the dpph ligand are estimated to be 73% and 0.83 ms, respectively. Thus, we successfully demonstrated a strong

photosensitized emission using the energy-donating ligand with a lower $T_1$ level than the emitting level of Tb(III).

**Mechanistic study.** To understand this characteristic energy migration system, we evaluated the photophysical properties of the **Tb-dpph** excited by dpph ligand under the presence of oxygen. An excited state equilibrium between Tb(III) and ligand $T_1$ was revealed through the emission lifetime measurements based on the oxygen concentrations[20,22–26] (Fig. S10, Ar: 0.83 ms, Air: 0.57 ms). The photosensitized emission quantum yield was also dependent on the oxygen concentrations (Ar: 73%, Air: 57%). The energy diagram for **Tb-dpph** is shown in Fig. 4b. From the fluorescence measurements (Fig. S11), the $S_1$ level of the dpph ligand (27,100 cm$^{-1}$) is lower than that of the tmh ligand (30,400 cm$^{-1}$). These results demonstrate that the effective photosensitized energy transfer occurs via the $T_1$ state of the dpph moiety in the Tb(III) complex. The $T_1$ level of the dpph ligand (19,850 cm$^{-1}$) is much lower than that of the tmh ligand (24,400 cm$^{-1}$)[37], hence indicating that the energy transfer pathway from the dpph to the tmh ligand is negligible. To further understand the excited state dynamics, we evaluated the temperature dependence of the emission intensity and 4f–4f emission lifetimes (Supplementary Notes 5, 6 and Figs. S12–18). The photosensitized emission intensity increased with the temperature from 100 to 400 K, suggesting the existence of a thermally-enhanced photosensitization pathway such as intersystem crossing[38] and/or energy transfer from $T_1$. The temperature-dependent emission measurement by direct 4f-4f excitation revealed the existence of a thermally-enhanced emission via the $T_1$ state in the excited-state equilibrium. The results suggest the existence of an endothermic energy transfer pathway corresponding to the $^7F_6 \rightarrow {}^5D_4$ transitions (Supplementary Note 7, Fig. S19, and Table S3). However, time-resolved emission spectroscopy showed a temperature-insensitive emission lifetime (100–350 K) at the excited-state equilibrium with the long-lived excited state of the dpph ligand (Supplementary Note 6 and

Figs. S16–18). The results suggest unusually efficient exothermic energy transfer pathways corresponding to the $^7F_5 \rightarrow {}^5D_4$ transitions from the $T_1$ states $(+\Delta E = 1400 \ cm^{-1})$ besides the endothermic energy transfer pathways corresponding to the $^7F_6 \rightarrow {}^5D_4$ transitions from the $T_1$ states $(-\Delta E = 650 \ cm^{-1})$. Theoretical studies suggest significantly populated $^7F_5$ owing to the long decay lifetime of $^7F_5 \rightarrow {}^7F_6$ in a relatively large energy gap between them (ca. $2050 \ cm^{-1}$)[39,40], allowing energy transfer from $^7F_5$ level[41]. Theoretical studies also indicate a larger energy-transfer matrix element for the $^7F_5 \rightarrow {}^5D_4$ transition than that for the $^7F_6 \rightarrow {}^5D_4$ transition[42]. The energy transfer from the $^7F_5$ state is one of the models for explaining the present temperature-insensitive lifetime behavior (the detailed discussion in Supplementary Note 6). Considering the temperature-dependent photophysical measurements and theoretical aspects, the characteristic thermally-assisted photosensitized emission occurs via the dpph $T_1$ state. Although determining the exact photosensitization pathway is difficult, this is, to the best of our knowledge, the first example of efficient photosensitized emission via the $T_1$ state in a lanthanide complex with an organic ligand $T_1$ level lower than the emitting level of the Ln(III) ion (Supplementary Note 8 and Figs. S20, S21).

## Conclusions

In this study, an effective photosensitized emission in a luminescent lanthanide complex with a $T_1$ level of an organic ligand lower than the emitting level of an Ln(III) ion was demonstrated. The thermally-assisted photosensitized emission was based on the excited-state equilibrium between a luminescent Ln(III) ion and an organic ligand with a persistent excited state. The photosensitizer model with a low $T_1$ level is advantageous for the construction of low-energy-driven photosensitization (Supplementary Note 9 and Fig. S22). The present study not only breaks the historical photosensitizer design rule based on Latva's rule, but also presents a novel photosensitizer model for photofunctional materials beyond lanthanide photochemistry.

## Methods

**General methods.** $^1H$ NMR spectrum was recorded in chloroform-d on a JEOL ECS-400 (400 MHz) spectrometer; TMS ($\delta_H = 0$ ppm) was used as the internal standard. Electrospray ionization (ESI) mass spectrometry were performed using the JEOL JMS-T100 LP instrument. Elemental analyses were performed using MICRO CORDER JM10. Emission spectra ($\lambda_{ex} = 356$ nm), excitation spectra ($\lambda_{em} = 548$ nm), and emission lifetimes ($\lambda_{ex} = 356$ nm and $\lambda_{em} = 548$ nm) for [Tb$_2$(tmh)$_6$(dpph)] were measured using a Horiba FluoroLog®3 spectrofluorometer. Temperature-dependent emission spectra for [Tb$_2$(tmh)$_6$(dpph)] ($\lambda_{ex} = 356$ nm and $\lambda_{ex} = 482$ nm) were measured using a Horiba FluoroLog®3 spectrofluorometer with a cryostat (Thermal Block Company SA-SB245T) and a temperature controller (Scientific Instruments Model 9700). Temperature-dependent emission lifetimes for [Tb$_2$(tmh)$_6$(dpph)] were measured using the third harmonics ($\lambda_{ex} = 355$ nm) of a Q-switched Nd:YAG laser (Spectra Physics, INDI-50, fwhm = 5 ns, $\lambda = 1064$ nm) and a photomultiplier (Hamamatsu Photonics, R5108, response time ≤1.1 ns) with a cryostat (Thermal Block Company SA-SB245T) and a temperature controller (Scientific Instruments Model 9700). The Nd:YAG laser response was monitored with a digital oscilloscope (Sony Tektronix, TDS3052, $f = 500$ MHz) synchronized to the single-pulse excitation. Emission quantum yields for [Tb$_2$(tmh)$_6$(dpph)] ($\lambda_{ex} = 370$ nm) and [Lu$_2$(tmh)$_6$(dpph)] ($\lambda_{ex} = 400$ nm) were measured using FP-6300 spectrofluorometer with an integration sphere (ILF-533). Emission spectrum ($\lambda_{ex} = 400$ nm, 100 K, delay: 80 msec) and time-resolved emission intensity ($\lambda_{ex} = 400$ nm, $\lambda_{em} = 530$ nm, 293 K) for [Lu$_2$(tmh)$_6$(dpph)] were measured using FP-6300 spectrofluorometer with a cryostat (Thermal Block Company SA-SB245T) and a temperature controller (Scientific Instruments Model 9700). Emission lifetimes were estimated using triple exponential functions in the region from 0.02 to 10 s based on the time-delayed-dependent emission spectral results (Fig. S8). The percentage of emission intensity contribution (~0 s) was calculated using the estimated triple exponential function. Emission images of [Lu$_2$(tmh)$_6$(dpph)] were taken by a camera (PENTAX, K-70).

**Materials.** Lutetium(III) nitrate hydrate (99.999%) was purchased from Aldrich Co., Ltd. Terbium(III) chloride hexahydrate (99.95%), $n$-butyllithium in $n$-hexane (1.6 mol/L), and chloroform-d (99.8%) were purchased from Kanto Chemical Co., Inc. Tetrahydrofuran, super dehydrated, with a stabilizer (for organic synthesis), and hydrogen peroxide (30%), sodium sulfate, anhydrous were purchased from

Wako Pure Chemical Industries, Ltd. 2,2,6,6-Tetramethyl-3,5-heptanedione (>97%), 2,7-dibromophenanthrene (>98.0%) and chlorodiphenylphosphine (>97.0%) were purchased from Tokyo Chemical Industry Co., Ltd.

**Preparation of [2,7-bis(diphenylphosphoryl)phenanthrene (dpph)].** A solution of $n$-butyllithium (6.2 mL, 9.9 mmol) was added dropwise to a solution of 2,7-dibromophenanthrene (1.67 g, 4.97 mmol) in dry tetrahydrofuran (45 mL) at –76 °C under Ar atmosphere. After 2 h, chlorodiphenylphosphine (1.8 mL, 9.8 mmol) was added to the solution at –76 °C under Ar atmosphere, and then stirred for 20 h at room temperature. The reaction mixture was added to dichloromethane, washed with water, and then dried over anhydrous sodium sulfate. The obtained solution was evaporated and chloroform (30 ml) was added to the product. A 30% hydrogen peroxide aqueous solution (4 mL) was added to the solution, and the reaction mixture was stirred for 2 h. The product was extracted using dichloromethane, and the extract was washed with water and then dried over anhydrous sodium sulfate. The compounds were purified by silica gel column chromatography (ethyl acetate: methanol = 23: 2) (Yield: 63.8%, 1.83 g, 3.16 mmol).
$^1H$ NMR (400 MHz, chloroform-d) $\delta$/ppm = 8.75 (dd, $J = 8.8$ Hz, 2.4 Hz, 2H), 8.31 (dd, $J = 13.4$ Hz, 1.0 Hz, 2H), 7.88 (t, $J = 9$ Hz, 2H), 7.79–7.47 (m, 22H); ESI-MS: $m/z$ calcd. for $[C_{38}H_{29}O_2P_2]^+ = 579.16$; found: 579.16. Elemental analysis calcd. (%) for $C_{38}H_{28}O_2P_2$, C 78.88, H 4.88; found: C 78.11, H 4.98.

**Preparation of [Tb$_2$(tmh)$_6$(dpph)].** Methanol solution (6 mL) containing Tb$_2$(tmh)$_6$ (99.2 mg, 0.07 mmol) and dpph (40.5 mg, 0.07 mmol) was refluxed for 18 h. The solution was filtered, and recrystallization from the solution gave white crystals (Yield: 41.1%, 57.5 mg, 0.0288 mmol).
ESI-MS: $m/z$ calcd. for $[C_{82}H_{104}O_{10}P_2Tb_2]^{2+} = 814.28$; found: 814.29. Elemental analysis calcd. (%) for $C_{104}H_{142}Tb_2O_{14}P_2$, C 62.58, H 7.17; found: C 62.25, H 7.14; IR(ATR) = 2961 (st, C-H), 1575 (st, C=O), 1183 $cm^{-1}$ (st, P=O).

**Preparation of [Lu$_2$(tmh)$_6$(dpph)].** Methanol solution (5 mL) containing Lu(tmh)$_3$ (101.6 mg, 0.14 mmol) and dpph (40.6 mg, 0.07 mmol) was refluxed for 16 h. The solution was filtrated, and recrystallization from the solution gave white crystals (Yield: 14.1%, 20.1 mg, 0.0099 mmol).
ESI-MS: $m/z$ calcd. for $[C_{82}H_{104}Lu_2O_{10}P_2]^{2+} = 830.29$; found: 830.28, Elemental analysis calcd. (%) $C_{104}H_{142}Lu_2O_{14}P_2$, C 61.59, H 7.06; found: C 61.19, H 7.06.

**Single-crystal X-ray structure determination.** X-ray crystal structures for [Tb$_2$(tmh)$_6$(dpph)] and [Lu$_2$(tmh)$_6$(dpph)] are shown in Fig. 2 and Fig. S2, respectively. The crystallographic data are shown in Table S1. Single crystal X-ray diffraction data were obtained using Rigaku XtaLAB Synergy-DW equipped with a HyPix-6000HE detector (MoK$_\alpha$ radiation, $\lambda = 0.71073$ Å). Non-hydrogen atoms were refined anisotropically using the SHELX system. Hydrogen atoms were refined using the riding model. All calculations were performed using the crystal structure crystallographic and Olex 2 software package. The CIF data were confirmed by the check CIF/PLATON service.

## Data availability

The single-crystal data generated in this study have been deposited in The Cambridge Crystallographic Data Center under accession code CCDC-2128731 (for [Tb$_2$(tmh)$_6$(dpph)], Supplementary Data 1) and CCDC-2128735 (for [Lu$_2$(tmh)$_6$(dpph)], Supplementary Data 2). These data can be obtained free of charge from The Cambridge Crystallographic Data Center via www.ccdc.cam.ac.uk/data_request/cif. All of the other data supporting the findings of this study are available from the corresponding author upon reasonable request.

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

## Acknowledgements

This work was partially supported by a grant-in-aid for grant numbers JP20H02748, JP20H04653, JP20H05197, JP20K21201, and JP21K18969. This research was supported by the Adaptable and Seamless Technology Transfer Program through Target-driven R&D (A-STEP: JPMJTM20J8) from Japan Science and Technology Agency (JST). This work was also supported by the Institute for Chemical Reaction Design and Discovery (ICReDD), established by the World Premier International Research Center Initiative (WPI) of MEXT, Japan.

## Author contributions

Y.K. designed research. K.S., T.N., and M.K. performed syntheses. K.S. and P.P.F.d.R. performed X-ray crystal measurements. K.S., M.K., and T.N. performed optical measurements. T.N. calculated the electronic structure of a lutetium complex. S.O. performed a simulation of the excited state dynamics. Y.K., K.S., T.N., M.K., S.O., P.P.F.d.R., S.S., K.F., and Y.H. wrote the paper. All authors reviewed the paper.

## Competing interests

The authors declare no competing interests.
