## [Peer Review File · Communications Chemistry]

Reviewers' comments:

Reviewer #1 (Remarks to the Author):

This is another interesting manuscript from Hasegawa's group where the authors well-conducted the synthesis and spectroscopic measurements of a novel and very curious dinuclear lanthanide complex [Tb₂(tmh)₆dpph]. My main concern is about the energy transfer process. With all respect, I disagree with the authors that it has an endothermic characteristic. So, I hope the authors receive my comments as an opportunity to improve even more the manuscript that, for sure, has merit to be published in *Communications Chemistry*.

From the phosphorescence spectrum (Figure 3a), the T₁ of dpph ligand is at 19850 cm⁻¹ and this should lie below the Tb(III) ⁵D₄ level in the diagram in Figure 4b. However, when referring that the ⁵D₄ level is at 20500 cm⁻¹, this energy is in respect to the ground ⁷F₆ level. Thus, the non-radiative energy transfer pathway that the authors are considering in the manuscript is described by the T₁→S₀ relaxation (19850 cm⁻¹) and rising of Tb(III) [⁷F₆ → ⁵D₄] (20500 cm⁻¹) with a negative donor-acceptor energy difference (ΔE = -650 cm⁻¹).

If the authors consider that the energy could be transferred to the Tb(III) ⁵D₄ level from a different pathway as from the Tb(III) ⁷F₅ level as a starting level? So, this pathway can be described as dpph [T₁→S₀] → Tb(III) [⁷F₅→⁵D₄] with a donor-acceptor energy difference of +1400 cm⁻¹, which is an exothermic energy transfer. To make this more clear, see Figure R1 below.

Figure R1. Simplified energy level diagram for the dpph to Tb(III) energy transfer.

The 7F_5 is not thermally coupled with the 7F_6 because of the large energy gap between them (ca. 2050 cm^{-1}), but it has a significant population due to the long decay lifetime of ${}^7F_5 \rightarrow {}^7F_6$ which could assume several ms, as measured in [10.1364/JOSAB.21.002117] and [10.1063/1.1901815]. Thus, the population of the 7F_5 level is not negligible and can receive energy in the energy transfer process [10.1016/j.jlumin.2022.118933]. This favors, even more, the energy transfer process because the ${}^7F_5 \rightarrow {}^5D_4$ transition involves higher matrix elements than the ${}^7F_6 \rightarrow {}^5D_4$ one [10.1002/adts.202000304].

Kasprzycka et al. [10.1016/j.jre.2020.02.001] reported an experimental value of the emission quantum yield of 58% for the $\text{Na}[\text{Tb}(\text{L})_4]$ (L = dimethyl (4-methylphenylsulfonyl) amidophosphate) compound. When the intramolecular energy transfer rates involving the 7F_5 level are excluded in theoretical simulations, the calculated emission quantum yield drops to 1.5%. Thus, I'm sure that this work from Hasegawa's group can bring high citations because provides great evidence of the 7F_5 level participation in the intramolecular energy transfer.

Minor comments and corrections:

Introduction, page 3:

“...which is mitigated by photosensitized energy transfer from organic compounds with a larger absorption coefficient...” change compounds to ligands

Introduction, page 4:

“Theoretical calculations have suggested that the T1-Ln(III) energy transfer rate is much higher than the lifetime of the excited states of Ln(III) ions.¹⁹” change lifetime (units of s) to “inverse of the lifetime” (units of s⁻¹ that is the same as energy transfer rate)

Experimental, page 9:

“X-ray crystal structures for [Tb₂(tmh)₆dpph] and [Lu₂(tmh)₆dpph] are shown in Figure 2 and Figure S1, respectively” correct respectively.

Results and discussion, page 10:

“To the best of our knowledge, this is the first occurrence of persistent luminescence (sub-second order) of organic ligands in lanthanide complexes at room temperature conditions (298 K).”

There are other examples of the appearance of the Ligand phosphorescence in Ln-complexes at room temperature.

In [10.1039/d1ce00228g], the measured phosphorescence lifetimes also have a very long decay time (> 100 ms). In [10.1016/j.tsf.2012.09.085], the author also detected the ligand phosphorescence at room temperature.

Results and discussion, page 12:

“which is ascribed to the exothermic energy transfer pathways from dpph S₁ to Tb(III) ⁵D₃ state”

S₁ to Tb(III) ⁵G₆ state is close to the ⁵D₃ and has a more relevant role in the intramolecular energy transfer. [10.1002/adts.202000304]

“The emission intensity is increased as increase in temperature from 100 to 300 K, indicating the existence of endothermic energy transfer pathway for the photosensitized emission.”

Probably, this is more related to the increase of the S₁-T₁ ISC rate rising with the temperature.

Figure S4: Please, insert the color labels in the figure caption.

Reviewer #2 (Remarks to the Author):

In this manuscript, authors demonstrate an efficient emission from a crystalline lanthanide complex using facilitated energy transfer owing to long-lived triplet state of molecular ligand. Two kinds of molecules, Ln-dpph and Tb-dpph, are synthesized and photophysical properties regarding their crystals are measured. They have comparable crystalline structure to allow a discussion of comparable packing condition of the ligand. Persistent emission characteristics with the average lifetime of 0.13 s of Ln-dpph crystals visualize the suppressed triplet deactivation of dpph ligand in crystalline condition. The long-lived triplet state of the ligand due to the suppressed triplet deactivation allowed the efficient endothermic energy transfer from the dpph ligand to Tb center. This contributed to the large emission yield of 73% as well as the emission lifetime of 0.83 ms for Tb-dpph crystals.

The two crystals are well designed to discuss the photophysical mechanism resulting in emission characteristics. This manuscript contains original points in the field relating lanthanide complexes and most information regarding reproducibility is appropriately explained.

However, some explanations still confuse readers in a variety of fields and there are still unclear points about discussions regarding a main critical point allowing the efficient endothermic energy transfer and emission from Tb center. Therefore, reviewer considers providing a comment about recommendation after receiving appropriate response to the following points in their revised manuscript.

(Comment and question 1)

Expressions in current abstract potentially give readers a misleading information that Tb-dpph crystals has the emission lifetime of 0.13 s as well as the emission yield of 73%. However, Tb-dpph crystals has the emission yield of 73% while the average emission lifetime is likely 0.83 ms. Therefore, more appropriate correction to minimize the misleading is necessary.

(Comment and Question 2)

Lines 1-3 in p.11 "The average τ of Tb(III) complexes" also potentially confuses readers to well understand the emission lifetime of Tb-dpph crystal. This is because readers may see Tb-dpph has the lifetime of emission with 130 ms. I now understand that the triplet level of Ln is large to deaccelerate the endothermic energy transfer from dpph ligand to the Ln center while the slow endothermic energy transfer is possible to allow the persistent emission because of the suppressed triplet deactivation of the dpph ligand. Although I could not well see the triplet energy level of Ln center and potentially have a wrong understanding, because the current sentences at least confuse readers, appropriate improvement to explain the point is necessary.

(Comment and Question 3)

Authors explain that energy levels are determined using band deconvolution analysis in Figure 3a. I could not well read how authors concretely determine the T1 energy level from the fitting data. More kind explanations about three fitting lines are recommended.

(Comment and Question 4)

Generally, exothermic process is much faster than endothermic process while authors need the endothermic process to obtain the better emitting performance. Therefore, there are other reasons to need the inefficient endothermic pathways. Because readers are difficult to see the point from current explanation in manuscript, kinder explanations are necessary for authors to see the critical point.

(Comment and Question 5)

If a decrease of S1 energy level is also crucial point for sensitization of Tb and the decrease also causes much a decrease of T1 energy of the ligand, long-lived triplet state of the ligand may be an intrinsic key. However, discussion why the long-lived triplet state of ligand could be observed for dpph

ligand is missing. Generally, various conjugated structures with triplet $n-n^*$ characteristics lose the long-lived characteristics in their crystalline structures, and there have been discussed before. Because authors show cif. files in supporting information, the discussion regarding reasons to allow the long-lived triplet state in dpph in crystalline state is necessary by referring appropriate references.

(Comment and question 6)

Authors show a schematic diagram to explain processes after excitation for Tb-dpph in Figure 4b. Readers cannot well image each rate of processes after the T1 generation of dpph ligand. Readers would like to see the deactivation rate from T1 in dpph ligand, the rate of endothermic energy transfer from T1 of dpph ligand to 5D4 of lanthanide center, and the rate of emission from 5D4 of lanthanide center for Ln-dpph crystal as well as Tb-dpph crystals. The similar kinetics including endothermic process have been well evaluated in the field of thermally activated delayed fluorescence, such characteristics helps readers to well and accurately understand the kinetics of excited state of two crystals based on the concrete values.

(Comment and question 7)

How about emission yield of Ln-dpph crystal? How about emission spectral intensity at 1 ms, 10 ms, 100 ms, and 1s after stopping excitation without normalizing the spectral intensity for Ln-dpph crystal? This is important for readers to image performance of persistent emission. Current information in Figure 3c is not kind for audience to see information of decay time less than 100 ms.

(Comment and question 8)

For example, how do emission characteristics of Ln-dpph in vacuum when it is doped with amorphous solid host with large T1 energy such as ZEONEX? Persistent emission caused by a rapid emission after endothermic process have been reported in thermally activated delayed fluorescence with the S1-T1 energy gap as well as long-lived T1 state when they are often doped in solid host with large T1 energy. If Ln-dpph molecule potentially has different points, they also will interest researchers in a variety of fields including organic opto-electronics fields.

(Comment and question 9)

In Figure S6, why emission intensity decreases at temperature more than 300K?

(Comment and question 10)

Authors see thermally activated signal of emission in Figure S6. How about the estimated activation energy of the endothermic energy transfer? The activation energy appropriately explains the energy difference (650 cm^{-1})?

(Comment and question 11)

Because readers working in other fields are not familiar with the emission yield of other lanthanide complexes, the sound of the emission yield of 73% in Tb-dpph crystal should be compared with previously reported other materials using lanthanide complexes.

Responses to the referee's comments and changes made in the revised manuscript

First, we would like to thank the referees and the editor for their valuable comments, based on which, we have revised our paper carefully.

Reviewer #1 (Remarks to the Author):

1. This is another interesting manuscript from Hasegawa's group where the authors well-conducted the synthesis and spectroscopic measurements of a novel and very curious dinuclear lanthanide complex [Tb₂(tmh)₆dp₂ph]. My main concern is about the energy transfer process. With all respect, I disagree with the authors that it has an endothermic characteristic. So, I hope the authors receive my comments as an opportunity to improve even more the manuscript that, for sure, has merit to be published in Communications Chemistry. Manuscript ID: COMMSCHEM-22-0148

From the phosphorescence spectrum (Figure 3a), the T₁ of dp₂ph ligand is at 19850 cm⁻¹ and this should lie below the Tb(III) ⁵D₄ level in the diagram in Figure 4b. However, when referring that the ⁵D₄ level is at 20500 cm⁻¹, this energy is in respect to the ground ⁷F₆ level. Thus, the non-radiative energy transfer pathway that the authors are considering in the manuscript is described by the T₁→S₀ relaxation (19850 cm⁻¹) and rising of Tb(III) [⁷F₆ → ⁵D₄] (20500 cm⁻¹) with a negative donor-acceptor energy difference (ΔE = -650cm⁻¹). If the authors consider that the energy could be transferred to the Tb(III) ⁵D₄ level from a different pathway as from the Tb(III) ⁷F₅ level as a starting level? So, this pathway can be described as dp₂ph [T₁→S₀] → Tb(III) [⁷F₅→⁵D₄] with a donor-acceptor energy difference of +1400 cm⁻¹, which is an exothermic energy transfer. To make this more clear, see Figure R1 below. The ⁷F₅ is not thermally coupled with the ⁷F₆ because of the large energy gap between them (ca. 2050 cm⁻¹), but it has a significant population due to the long decay lifetime of ⁷F₅→⁷F₆ which could assume several ms, as measured in [10.1364/JOSAB.21.002117] and [10.1063/1.1901815]. Thus, the population of the ⁷F₅ level is not negligible and can receive energy in the energy transfer process [10.1016/j.jlumin.2022.118933]. This favors, even more, the energy transfer process because the ⁷F₅→⁵D₄ transition involves higher matrix elements than the ⁷F₆→⁵D₄ one [10.1002/adts.202000304]. Kasprzycka et al. [10.1016/j.jre.2020.02.001] reported an experimental value of the emission quantum yield of 58% for the Na[Tb(L)₄] (L = dimethyl (4-methylphenylsulfonyl) amidophosphate) compound. When the intramolecular energy transfer rates involving the ⁷F₅ level are excluded in theoretical simulations, the calculated emission quantum yield

drops to 1.5%. Thus, I'm sure that this work from Hasegawa's group can bring high citations because provides great evidence of the 7F_5 level participation in the intramolecular energy transfer.

<Answer 1> We agree with reviewer's comment. The oxygen-dependent emission lifetimes indicate an excited-state equilibration between 5D_4 and T_1 . To understand the mechanism further, we conducted temperature-dependent emission lifetime analyses (100–400 K). The emission lifetime was almost unchanged in this region ($\tau < 1$ ms). Thus, a long 4f–4f emission lifetime was not observed in the Tb(III) complex, despite a significantly long T_1 lifetime of the dppe moiety. These results indicated the existence of other effective pathways for an exothermic energy transfer from the T_1 states. We also believe that these energy-transfer pathways are related to the ${}^7F_5 \rightarrow {}^5D_4$ transition. We have modified the manuscript accordingly and added the temperature-dependent emission lifetimes in the Supporting Information.

<Manuscript, P12–P13>

To further understand the excited state dynamics, we evaluated the temperature dependence of the photosensitized emission intensity (Supplementary Note 2, Figures S8–S9) and their emission lifetimes (Supplementary Note 3, Figure S10-11). An effective photosensitized Tb(III) emission was observed in the low-temperature region (100 K), which was ascribed to the exothermic energy transfer from dppe S_1 to Tb(III) 5G_6 and 5D_3 . The emission intensity increased with the temperature from 100 to 300 K, indicating the existence of a thermally enhanced photosensitization pathway such as intersystem crossing³⁹ and/or energy transfer from T_1 . Time-resolved emission spectroscopy revealed a temperature-insensitive emission lifetime at the excited-state equilibrium with the long-lived excited state of the dppe ligand (Supplementary Note 3, Figure S10). The results indicate unusually efficient exothermic energy transfer pathways corresponding to the ${}^7F_5 \rightarrow {}^5D_4$ transitions from the T_1 states ($+\Delta E = 1,400$ cm^{-1}) besides the endothermic energy transfer pathways corresponding to the ${}^7F_6 \rightarrow {}^5D_4$ transitions from the T_1 states ($-\Delta E = 650$ cm^{-1}). Theoretical studies also indicate a significantly populated 7F_5 owing to the long decay lifetime of ${}^7F_5 \rightarrow {}^7F_6$ in a relatively large energy gap between them (ca. 2,050 cm^{-1}),⁴⁰⁻⁴¹ allowing energy transfer from 7F_5 level.⁴² Theoretical studies also indicate a larger energy-transfer matrix element for the ${}^7F_5 \rightarrow {}^5D_4$ transition than that for the ${}^7F_6 \rightarrow {}^5D_4$ transition.⁴² Considering the temperature-dependent photophysical measurements and theoretical aspects, the characteristic thermally assisted energy transfer occurs from the dppe T_1 to the Tb(III) 5D_4 state (Figure 4b). This is the first demonstration of an efficient photosensitized emission of a lanthanide complex with a T_1 level of an organic ligand lower than the emitting level of a Ln(III) ion.

39. Rademaker, K., Krupke, W. F., Page, R. H., Payne, S. A., Petermann, K., Huber, G., Yelisseyev, A. P., Isaenko, L. I., Roy, U. N., Burger, A., Mandal, K. C., and Nitsch, K. Optical properties of Nd³⁺ and Tb³⁺-doped KPb₂Br₅ and RbPb₂Br₅ with low nonradiative decay. *J. Opt. Soc. Am. B* **21**, 2117 (2004).
40. Roy, U. N., Hawrami, R. H., Cui, Y., Morgan, S., Burger, A., Mandal, K. C., Noblitt, C. C., Speakman, S. A., Rademaker, K., Payne, S. A. Tb³⁺-doped KPb₂Br₅: Low-energy phonon mid-infrared laser crystal. *Appl. Phys. Lett.* **86**, 151911 (2005).
41. Carneiro Neto, A. N., Kasprzycka, E., Souza, A. S., Gawryszewska, P., Suta, M., Carlos, L. D., Malta, O. L. On the long decay time of the ⁷F₅ level of Tb³⁺. *J. Lumin.* **248**, 118933 (2022).
42. Moura, R. T., Oliveira, J. A., Santos, I. A., de Lima, E. M., Carlos, L. D., Aguiar, E. C., Neto, A. N. C., Theoretical evidence of the singlet predominance in the intramolecular energy transfer in ruhemann's purple Tb(III) complexes. *Adv. Theory Simul.* **4**, 2000304 (2021)

<Supporting Information, P7-P9>

Supplementary Note 3. Temperature-dependent emission lifetimes for Tb-dpph and their comparison

We measured the temperature-dependent emission lifetimes using **Tb-dpph** (Figure S10). The other data (Figure S11, [Tb(tmh)₃(tppo)]^{S3} (tppo: triphenylphosphine oxide) and [Tb₂(tmh)₆(bpeb)] (bpeb: 1,4-bis(diphenylphosphoryl)ethynylbenzene)) are also shown in Figure S10 for comparison. The T₁ level of the extended π -conjugated bpeb ligand is estimated to be 21,230 cm⁻¹, from TD-DFT calculations (B3LYP/6-31G(D)) using the structure optimized by DFT calculations (B3LYP/6-31G(D)). The emission lifetimes of [Tb(tmh)₃(tppo)] and [Tb₂(tmh)₆(bpeb)], with the extended π -conjugated systems, are insensitive and sensitive to temperature, respectively. The temperature-dependent emission lifetime of [Tb₂(tmh)₆(bpeb)] can be attributed to the deactivation pathway induced by back energy transfer from ⁵D₄ to T₁.^{S4} In contrast, **Tb-dpph** has a large π -conjugated phenanthrene framework with a low T₁ level (19,850 cm⁻¹), and its emission lifetime is insensitive to temperature. In the excited dynamics system, an excited-state equilibrium between the T₁ and ⁵D₄ levels was revealed by the oxygen-dependent emission lifetime. These results indicate the existence of a mechanism for suppression of the quenching effect via the T₁ state with an excited-state equilibrium. A long T₁ lifetime is expected to allow an efficient use of photons, even in the case of a low T₁ level, for an excited-state equilibration between T₁ and ⁵D₄. In general, an excited-state equilibrium, with an energy-donating state having a lower energy level than the energy-accepting state and a significantly slower deactivation rate to the ground state than that from the accepting state to the ground state, should provide a significantly longer lifetime;^{S5} however, the emission lifetime

of **Tb-dpph** is just slightly longer than that of other Tb(III) complexes containing tmh and phosphine oxide ligands (Figure S10). These results indicate an unusually efficient exothermic energy transfer corresponding to the ${}^7F_5 \rightarrow {}^5D_4$ transition from the T_1 states, besides the endothermic energy transfer corresponding to the ${}^7F_6 \rightarrow {}^5D_4$ transition (Figure 4b; the energy levels have been summarized in Table S2). 7F_5 has a significant population, owing to a long decay lifetime of ${}^7F_5 \rightarrow {}^7F_6$ in a relative large energy gap between them ($2,050 \text{ cm}^{-1}$),^{S6-S9} allowing energy transfer from the 7F_5 level.^{S10} Theoretical calculations also suggest a larger matrix element for the energy-transfer rate of the ${}^7F_5 \rightarrow {}^5D_4$ transition than that of the ${}^7F_6 \rightarrow {}^5D_4$ transition.^{S11} Considering the experimental and theoretical aspects, the characteristic thermally assisted energy transfer occurs from the dpph T_1 to the Tb(III) 5D_4 state.

Figure S10. Temperature-dependent emission lifetimes for **Tb-dpph** (black dot, $\lambda_{\text{ex}} = 356 \text{ nm}$, $\lambda_{\text{em}} = 548 \text{ nm}$), $[\text{Tb}(\text{tmh})_3(\text{tppo})]$ (red dot, $\lambda_{\text{ex}} = 355 \text{ nm}$, $\lambda_{\text{em}} = 540 \text{ nm}$), and $[\text{Tb}_2(\text{tmh})_6(\text{bpeb})]$ (blue dot, $\lambda_{\text{ex}} = 355 \text{ nm}$, $\lambda_{\text{em}} = 540 \text{ nm}$).

Figure S11. The chemical structures of (a) $[\text{Tb}(\text{tmh})_3(\text{tppo})]$ and (b) $[\text{Tb}_2(\text{tmh})_6(\text{bpeb})]$.

Table S2. Energy levels of ligands and Tb(III).

	Energy / cm ⁻¹
S ₁ energy (dpph)	27,100
T ₁ energy (dpph)	19,850
S ₁ energy (tmh)	30,400
T ₁ energy (tmh)	24,400
⁵ G ₆ (Tb)	26,510 (24,460) ^a
⁵ D ₃ (Tb)	26,320 (24,270) ^b
⁵ D ₄ (Tb)	20,500 (18,450) ^c

a: Energy gap between ⁵G₆ and ⁷F₅. b: Energy gap between ⁵D₃ and ⁷F₅. c: Energy gap between ⁵D₄ and ⁷F₅.

S3. Ferreira da Rosa, P. P. et al. Thermosensitive seven-coordinate Tb^{III} complexes with LLCT transitions. *Eur. J. Inorg. Chem.* **2018**, 2031 (2018).

S4. Latva, M., Takalo, H., Mikkala, V.-M., Matachescu, C., Rodríguez-Ubis, J. C., Kankare, J. Correlation between the lowest triplet state energy level of the ligand and lanthanide(III) luminescence quantum yield. *J. Lumin.* **75**, 149 (1997).

S5. McClenaghan, N. D., Leydet, Y., Maubert, B., Indelli, M. T., Campagna, S. Excited-state equilibration: A process leading to long-lived metal-to-ligand charge transfer luminescence in supramolecular systems. *Coord. Chem. Rev.* **249**, 1336 (2005).

S6. Rademaker, K. et al, Optical properties of Nd³⁺ and Tb³⁺-doped KPb₂Br₅ and RbPb₂Br₅ with low nonradiative decay. *J. Opt. Soc. Am. B* **21**, 2117 (2004).

S7. Roy, U. N. et al., Tb³⁺-doped KPb₂Br₅: Low-energy phonon mid-infrared laser crystal. *Appl. Phys. Lett.* **86**, 151911 (2005).

S8. Seven-coordinated lanthanide complexes with tmh and phosphine oxide ligands have provided a relatively small non-radiative rate constant in the emission process.^{S9} These types of structures might be key for construction of long-lived ⁷F₅ in Tb^{III} complexes with vibrational organic ligands.

S9. Yanagisawa, K. et al., Seven-coordinate luminophores: Brilliant luminescence of lanthanide complexes with C_{3v} geometrical structures. *Eur. J. Inorg. Chem.* **2015**, 4769 (2015).

S10. Carneiro Neto, A. N. et al., On the long decay time of the ⁷F₅ level of Tb³⁺. *J. Lumin.* **248**, 118933 (2022).

S11. Moura, R. T., Theoretical evidence of the singlet predominance in the intramolecular energy transfer in ruhemann's purple Tb(III) complexes. *Adv. Theory Simul.* **4**, 2000304 (2021)

2. Introduction, page 3: "...which is mitigated by photosensitized energy transfer from organic compounds with a larger absorption coefficient..." change compounds to ligands

<Answer 2> Thank you for your helpful comment. As per the referee's suggestion, we have modified the sentence as follows:

<Manuscript, P3> "However, they exhibit low absorption coefficients ($\epsilon = 0.1-10 \text{ M}^{-1} \text{ cm}^{-1}$), which is mitigated by photosensitized energy transfer from organic ligands with larger absorption coefficients ($\epsilon = 10^3-10^5 \text{ M}^{-1} \text{ cm}^{-1}$)."

3. Introduction, page 4: "Theoretical calculations have suggested that the T₁-Ln(III) energy transfer rate is much higher than the lifetime of the excited states of Ln(III) ions.¹⁹" change lifetime (units of s) to "inverse of the lifetime" (units of s⁻¹ that is the same as energy transfer rate)

<Answer 3> Thank you for your helpful comment. As per the referee's suggestion, we have modified the sentence as follows:

<Manuscript, P4> "Theoretical calculations have suggested that the T₁-Ln(III) energy transfer rate is significantly higher than the inverse of the lifetime of the excited states of the Ln(III) ions."

4. Experimental, page 9: "X-ray crystal structures for [Tb₂(tmh)₆dp₂ph] and [Lu₂(tmh)₆dp₂ph] are shown in Figure2 and Figure S1, respectively" correct respectively

<Answer 4> Thank you for your helpful comment. According to the referee's comment, we have modified the sentence as follows:

<Manuscript, P9> "The X-ray crystal structures for [Tb₂(tmh)₆(dp₂ph)] and [Lu₂(tmh)₆(dp₂ph)] are shown in Figures 2 and S1, respectively."

5. Results and discussion, page 10: "To the best of our knowledge, this is the first occurrence of persistent luminescence (sub-second order) of organic ligands in lanthanide complexes at room temperature conditions(298 K)." There are other examples of the appearance of the Ligand phosphorescence in Ln-complexes at room temperature. In [10.1039/d1ce00228g], the measured phosphorescence lifetimes also have a very long decay time (> 100 ms). In [10.1016/j.tsf.2012.09.085], the author also detected the ligand

phosphorescence at room temperature.

<Answer 5> We thank the referee for having pointed this out. However, the introduced references for the reported T_1 lifetimes in lanthanide complexes did not show the emission lifetimes in [10.1016/j.tsf.2012.09.085] and found them only for the low-temperature condition (200 K) in [10.1039/d1ce00228g]. We also re-took an emission image using high-resolution camera (PENTAX, K-70) and reanalyzed the time-resolved emission decay carefully. Consequently, we have modified the sentence as follows:

<Manuscript, P10> The emission photograph of **Lu-dpph** is shown in Figure 3b, where it shows a green persistent emission. The emission durability of **Lu-dpph** was evaluated using time-resolved emission spectroscopy (Figure 3c), yielding characteristic emission-decay curves for persistent-emission materials (Supplementary Note 1, Figure S5-S7). Herein, the emission lifetimes were estimated using triple exponential functions ($\tau_1 = 16$ ms (70 %), $\tau_2 = 83$ ms (27%), and $\tau_3 = 450$ ms (3%)). The average $\pi-\pi^*$ emission lifetime of the dpph ligand in **Lu-dpph** was estimated to be 47 ms, which is characteristic long among T_1 lifetime of organic ligands in lanthanide complexes at room temperature.³⁰⁻³³

Figure 3. The emission spectrum of **Lu-dpph** (a: $\lambda_{\text{ex}} = 400$ nm; delay: 80 ms; 100 K). Emission images of **Lu-dpph** excited using UV-light (b: $\lambda_{\text{ex}} = 375$ nm; 293 K) under vacuum condition. Emission decay curves of **Lu-dpph** (c: $\lambda_{\text{ex}} = 400$ nm; $\lambda_{\text{em}} = 530$ nm; 293 K).

30. Kalota, B., Tsvirko, M. Fluorescence and phosphorescence of lutetium(III) and gadolinium(III) porphyrins for the intraratiometric oxygen sensing. *Chem. Phys. Lett.* **634**, 188 (2015).
31. Sun, B., Wei, C., Wei, H., Cai, Z., Liu, H., Zang, Z., Yan, W., Liu, Z., Bian, Z., Huang, C. Highly efficient room-temperature phosphorescence achieved by gadolinium complexes. *Dalton Trans* **48**, 14958 (2019).
32. Zhao, Z. L., Ru, J. X., Zhou, P. P., Wang, Y. S., Shan, C. F., Yang, X. X., Cao, J., Liu, W. S., Guo, H. C., Tang, Y. A smart nanoprobe based on a gadolinium complex encapsulated by ZIF-8 with enhanced room temperature phosphorescence for synchronous oxygen sensing and photodynamic therapy. *Dalton Trans* 2019, **48**, 16952.
33. Kitagawa, Y., Moriake, R., Akama, T., Saito, K., Aikawa, K., Shoji, S., Fushimi, K., Kobayashi, M., Taketsugu, T., Hasegawa, Y. Effective photosensitization in excited-state equilibrium: Brilliant luminescence of Tb^{III} coordination polymers through ancillary ligand modifications. *ChemPlusChem*, e202200151 (2022).

6. Results and discussion, page 12: “which is ascribed to the exothermic energy transfer pathways from dpph S₁ to Tb(III) ⁵D₃ state” S₁ to Tb(III) ⁵G₆ state is close to the ⁵D₃ and has a more relevant role in the intramolecular energy transfer. [10.1002/adts.202000304]

<Answer 6> We appreciate the referee’s suggestion. We modified the sentence as follows:

<Manuscript, P12> An effective photosensitized Tb(III) emission was observed in the low-temperature region (100 K), which was ascribed to the exothermic energy transfer from dpph S₁ to Tb(III) ⁵G₆ and ⁵D₃.

7. “The emission intensity is increased as increase in temperature from 100 to 300 K, indicating the existence of endothermic energy transfer pathway for the photosensitized emission.” Probably, this is more related to the increase of the S1-T1 ISC rate rising with the temperature.

<Answer 7> Thank you for your helpful comment. We agree with the referee’s suggestion and have modified the sentence, and added the explanation in the Supporting Information, as follows:

<Manuscript, P 12–13>

“The emission intensity increased with the temperature from 100 to 300 K, indicating the existence of a thermally enhanced photosensitization pathway such as intersystem crossing³⁸ and/or energy transfer from T₁.”

<Supporting Information, P5>

“An increase in the emission intensity with temperature (100–300 K) indicates thermally enhanced photosensitized emission pathways such as intersystem crossing efficiency^{S1} or energy-transfer efficiency from T₁. A decrease in the emission intensity with increasing temperature (300 – 400 K) is considered to be originated from the increased non-radiative rate constant of T₁→S₀ transition in dpph ligand.

8. Figure S4: Please, insert the color labels in the figure caption

<Answer 8> As per the referee’s suggestion, we have modified the figure caption as follows:

<Supporting Information, P2>

Figure S4. Emission spectra of the dpph (black line: chloroform, 1.0×10^{-3} M, $\lambda_{\text{ex}} = 300$ nm) and tmh ligands in [Lu₂(tmh)₆] (red line: methanol, 1.0×10^{-4} M, $\lambda_{\text{ex}} = 230$ nm).

Reviewer #2 (Remarks to the Author):

General Comments

In this manuscript, authors demonstrate an efficient emission from a crystalline lanthanide complex using facilitated energy transfer owing to long-lived triplet state of molecular ligand. Two kinds of molecules, Ln-dpph and Tb-dpph, are synthesized and photophysical properties regarding their crystals are measured. They have comparable crystalline structure to allow a discussion of comparable packing condition of the ligand. Persistent emission characteristics with the average lifetime of 0.13 s of Ln-dpph crystals visualize the suppressed triplet deactivation of dpph ligand in crystalline condition. The long-lived triplet state of the ligand due to the suppressed triplet deactivation allowed the efficient endothermic energy transfer from the dpph ligand to Tb center. This contributed to the large emission yield of 73% as well as the emission lifetime of 0.83 ms for Tb-dpph crystals.

The two crystals are well designed to discuss the photophysical mechanism resulting in emission characteristics. This manuscript contains original points in the field relating lanthanide complexes and most information regarding reproducibility is appropriately explained.

However, some explanations still confuse readers in a variety of fields and there are still unclear points about discussions regarding a main critical point allowing the efficient endothermic energy transfer and emission from Tb center. Therefore, reviewer considers providing a comment about recommendation after receiving appropriate response to the following points in their revised manuscript.

<Answer> We appreciate the critical comments from reviewers 1 and 2 and have re-measured and reanalyzed various photophysical properties and re-evaluated our manuscript accordingly.

1. Expressions in current abstract potentially give readers a misleading information that Tb-dpph crystals has the emission lifetime of 0.13 s as well as the emission yield of 73%. However, Tb-dpph crystals has the emission yield of 73% while the average emission lifetime is likely 0.83 ms. Therefore, more appropriate correction to minimize the misleading is necessary.

<Answer 1> To clarify the information on the emission yield and lifetime, as per the reviewer's comment, we have modified the abstract as follows:

<Abstract>

Photosensitizer design strategy to allow efficient energy transfer has been studied for developing photofunctional molecular system. Herein, we demonstrate a novel photosensitizer design using dinuclear luminescent lanthanide complex, which exhibits photosensitized emission based on thermally-assisted energy transfer. The lanthanide complex comprised Tb(III) ions, six tetramethylheptanedionates, and phosphine oxide bridge containing a phenanthrene frameworks. The phenanthrene ligand and Tb(III) ions are the energy donor (photosensitizer) and acceptor (emission center) parts, respectively. The energy-donating level of the ligand (lowest excited triplet (T_1) level = $19,850\text{ cm}^{-1}$) is lower than the emitting level of the Tb(III) ion (5D_4 level = $20,500\text{ cm}^{-1}$). The long-lived T_1 state of the energy-donating ligands promoted an efficient thermally assisted energy transfer to the Tb(III) acceptor (5D_4 level), resulting in a pure-green colored emission with a high photosensitized emission quantum yield (73 %).

2. Lines 1-3 in p.11 “The average • • Tb(III) complexes” also potentially confuses readers to well understand the emission lifetime of Tb-dpph crystal. This is because readers may see Tb-dpph has the lifetime of emission with 130 ms. I now understand that the triplet level of Ln is large to decelerate the endothermic energy transfer from dpph ligand to the Ln center while the slow endothermic energy transfer is possible to allow the persistent emission because of the suppressed triplet deactivation of the dpph ligand. Although I could not well see the triplet energy level of Ln center and potentially have a wrong understanding, because the current sentences at least confuse readers, appropriate improvement to explain the point is necessary.

<Answer 2> Thank you for your suggestion. To avoid confusing the readers, as mentioned by the reviewer, we have re-considered the sentences in the manuscript. We also reanalyzed the time-resolved emission decay carefully. Consequently, we have also modified the manuscript as follows.

<Manuscript, P10–11>

The emission durability of **Lu-dpph** was evaluated using time-resolved emission spectroscopy (Figure 3c), yielding characteristic emission-decay curves for persistent-emission materials (Supplementary Note 1). Herein, the emission lifetimes were estimated using triple exponential functions ($\tau_1 = 16\text{ ms}$ (70 %), $\tau_2 = 83\text{ ms}$ (27%), and $\tau_3 = 450\text{ ms}$ (3%)). The average $\pi-\pi^*$ emission lifetime of the dpph ligand in **Lu-dpph** was estimated to be 47 ms, which is characteristic long among T_1 lifetime of organic ligands in lanthanide complexes at room temperature.^{30–33} The long T_1 lifetime in the dpph moiety was ascribed to

the rigid isolated polyaromatic structure encapsulated in the tmh ligands, which suppressed the non-radiative deactivation pathways.^{29,34-35} These results indicate the construction of an energy-donating system with a long T_1 lifetime in **Tb-dpph**. This dpph T_1 lifetime (47 ms) is significantly longer than the 4f–4f emission lifetimes of reported Tb(III) complexes.^{11,36}

29. Yang, X. G. Lu, X. M., Zhai, Z. M., Zhao, Y., Liu, X. Y., Ma, L. F., Zang, S. Q. Facile synthesis of micro-scale MOF host-guest with long-last phosphorescence and enhanced optoelectronic performance. *Chem. Commun.* **55**, 11099 (2019).

30. Kalota, B., Tsvirko, M. Fluorescence and phosphorescence of lutetium(III) and gadolinium(III) porphyrins for the intraratiometric oxygen sensing. *Chem. Phys. Lett.* **634**, 188 (2015).

31. Sun, B., Wei, C., Wei, H., Cai, Z., Liu, H., Zang, Z., Yan, W., Liu, Z., Bian, Z., Huang, C. Highly efficient room-temperature phosphorescence achieved by gadolinium complexes. *Dalton Trans* **48**, 14958 (2019).

32. Zhao, Z. L., Ru, J. X., Zhou, P. P., Wang, Y. S., Shan, C. F., Yang, X. X., Cao, J., Liu, W. S., Guo, H. C., Tang, Y. A smart nanoprobe based on a gadolinium complex encapsulated by ZIF–8 with enhanced room temperature phosphorescence for synchronous oxygen sensing and photodynamic therapy. *Dalton Trans* 2019, 48, 16952.

33. Kitagawa, Y., Moriake, R., Akama, T., Saito, K., Aikawa, K., Shoji, S., Fushimi, K., Kobayashi, M., Taketsugu, T., Hasegawa, Y. Effective photosensitization in excited-state equilibrium: Brilliant luminescence of Tb^{III} coordination polymers through ancillary ligand modifications. *ChemPlusChem*, e202200151 (2022).

34. Mieno, H., Kabe, R., Notsuka, N., Allendorf, M. D., Adachi, C. Long-lived room-temperature phosphorescence of coronene in zeolitic imidazolate framework ZIF-8. *Adv. Opt. Mater.* **4**, 1015–1021 (2016).

35. Hirata, S., Vacha, M. White Afterglow Room-Temperature Emission from an Isolated Single Aromatic Unit under Ambient Condition. *Adv. Opt. Mater.* **5**, 1600996 (2017).

36. Yanagisawa, K., Nakanishi, T., Kitagawa, Y., Seki, T., Akama, T., Kobayashi, M., Taketsugu, T., Ito, H., Fushimi, K., Hasegawa, Y. Seven-coordinate luminophores: Brilliant luminescence of lanthanide complexes with C_{3v} geometrical structures. *Eur. J. Inorg. Chem.* **2015**, 4769 (2015).

37. Yanagisawa, K., Kitagawa, Y., Nakanishi, T., Akama, T., Kobayashi, M., Seki, T., Fushimi, K., Ito, H., Taketsugu, T., Hasegawa, Y. Enhanced luminescence of asymmetrical seven-coordinate Eu^{III} complexes including LMCT perturbation. *Eur. J. Inorg. Chem.* **2017**, 3843 (2017).

3. Authors explain that energy levels are determined using band deconvolution analysis in Figure 3a. I could not well read how authors concretely determine the T₁ energy level from the fitting data. More kind explanations about three fitting lines are recommended.

<Answer 3> Thank you for your suggestion. To better explain the fitting data, as per the reviewer's comment, we have modified the following modifications in the manuscript:

<Manuscript P10>

The emission spectrum was deconvoluted into three vibronic bands using the software (OriginPro 2021b), the spectrum in wavenumber scale, and by fitting the peak profile using Gaussian functions (Figure 3a, broken line). The deconvolution results in the three vibronic bands were designated as 0-0 (19,850 cm⁻¹), 0-1 (18,670 cm⁻¹), and 0-2 (17,390 cm⁻¹). Thus, the T₁ level of the dp^{ph} ligand in **Lu-dp^{ph}** was determined to be 19,850 cm⁻¹ using band-deconvolution analysis.

4. Generally, exothermic process is much faster than endothermic process while authors need the endothermic process to obtain the better emitting performance. Therefore, there are other reasons to need the inefficient endothermic pathways. Because readers are difficult to see the point from current explanation in manuscript, kinder explanations are necessary for authors to see the critical point.

<Answer 4> Thank you for your comments. We have explained the advantage of photosensitized emission using a photosensitizer with low T₁ level, in the manuscript and supplementary information, as follows:

<Conclusion, P14>

In this study, an efficient photosensitized emission in a luminescent lanthanide complex with a T₁ level of an organic ligand lower than the emitting level of a Ln(III) ion was demonstrated for the first time. The thermally assisted energy transfer was based on the excited-state equilibrium between a luminescent Ln(III) ion and an organic ligand with a persistent excited state. The photosensitizer model with a low T₁ level is advantageous for the construction of low-energy driven photosensitization (Supplementary Note 5). The present study not only breaks the historical photosensitizer design rule based on the Latva's rule, but also presents a novel photosensitizer model for photofunctional materials beyond lanthanide photochemistry.

<Supporting Information, P12>

Supplementary Note 5

Advantages of photosensitization with a low T₁ level for enhancement of brightness

Tb(III) complexes with high emission quantum yields ($\geq 70\%$) have been already reported.^{S14-S19} The focus of this study was achieving highly efficient photosensitization using a donating ligand with a low T₁ level, which resulted in the excitation of the low S₁ level, resulting in an effective low-energy-driven photosensitization.^{S20} The energy-accepting level of the Tb^{III} ion was relatively high (⁵D₄: 20,500 cm⁻¹), requiring significantly high T₁ and S₁ level in the organic ligands for effective photosensitization. Brightness is defined by the product of molar absorption coefficient and emission quantum yield.^{S21} For example, [Tb(tmh)₃(tppo)] also produces a high emission quantum yield ($\Phi = 66\%$)^{S3} but its brightness, when excited using UV-light ($\lambda = 365$ nm), is estimated to be approximately 6 M⁻¹ cm⁻¹, based on the molar absorption coefficient (Figure S14a, $\epsilon_{365\text{nm}} = 9$ M⁻¹ cm⁻¹). In contrast, the brightness of **Tb-dpph** excited using UV-light ($\lambda = 365$ nm) was estimated to be 1,300 M⁻¹ cm⁻¹ (Figure S14b, $\epsilon_{365\text{nm}} = 1,780$ M⁻¹ cm⁻¹). Thus, our concept is expected to be useful for improving light-absorption properties and brightness.

Figure S14. The electronic absorption spectra of [Tb(tmh)₃(tppo)] in toluene (a, 5.0 × 10⁻³ M) and **Tb-dpph** in chloroform (b, 5.6 × 10⁻⁴ M).

S14. Kitagawa, Y. *et al.*, Effective photosensitization in excited-state equilibrium: Brilliant luminescence of Tb^{III} coordination polymers through ancillary ligand modifications. *ChemPlusChem*, e202200151 (2022).

S15. Correia, S. F. H. *et al.* High emission quantum yield Tb³⁺-activated organic-inorganic hybrids for

- UV-down-shifting green light-emitting diodes. *Eur. J. Inorg. Chem.* **2020**, 1736 (2020).
- S16. Chen, B.-L. *et al.* A thermostable terbium(III) complex with high fluorescence quantum yields. *New J. Chem.* **46**, 11021 (2022).
- S17. Bünzli, J.-C. G. On the design of highly luminescent lanthanide complexes. *Coord. Chem. Rev.* **293–294**, 19 (2015).
- S18. Xia, T. *et al.*, A terbium metal–organic framework for highly selective and sensitive luminescence sensing of Hg²⁺ ions in aqueous solution. *Chem.–Eur. J.* **22**, 18429 (2016).
- S19. Aquino, L. E. N. *et al.*, Seven-coordinate Tb³⁺ complexes with 90% quantum yields: high-performance examples of combined singlet- and triplet-to-Tb³⁺ energy-transfer pathways. *Inorg. Chem.* **60**, 892 (2021).
- S20. Kitagawa, Y. *et al.*, Stacked nanocarbon photosensitizer for efficient blue light excited Eu(III) emission. *Commun. Chem.* **3**, 3 (2020).
- S21. Wong, K.-L., Bünzli, J.-C. G., Tanner, P. A. Quantum yield and brightness. *J. Lumin.* **224**, 117256 (2020).

5. If a decrease of S₁ energy level is also crucial point for sensitization of Tb and the decrease also causes much a decrease of T₁ energy of the ligand, long-lived triplet state of the ligand may be an intrinsic key. However, discussion why the long-lived triplet state of ligand could be observed for dpph ligand is missing. Generally, various conjugated structures with triplet π-π* characteristics lose the long-lived characteristics in their crystalline structures, and there have been discussed before. Because authors show cif. files in supporting information, the discussion regarding reasons to allow the long-lived triplet state in dpph in crystalline state is necessary by referring appropriate references.

<Answer 5> Thank you for your helpful suggestion. To better explain the long-lived triplet state in the crystalline dpph, we have modified the manuscript, citing the appropriate references, as follows:

<Manuscript, P10>

The long T₁ lifetime in the dpph moiety was ascribed to the rigid isolated polyaromatic structure encapsulated in the tmh ligands, as revealed via crystal analysis, which suppressed the non-radiative deactivation pathways.^{29, 34-35}

29. Yang, X. G. Lu, X. M., Zhai, Z. M., Zhao, Y., Liu, X. Y., Ma, L. F., Zang, S. Q. Facile synthesis of micro-scale MOF host-guest with long-last phosphorescence and enhanced optoelectronic performance. *Chem. Commun.* **55**, 11099 (2019).
34. Mieno, H., Kabe, R., Notsuka, N., Allendorf, M. D., Adachi, C. Long-lived room-temperature phosphorescence of coronene in zeolitic imidazolate framework ZIF-8. *Adv. Opt. Mater.* **4**, 1015 (2016).
35. Hirata, S. & Vacha, M. White Afterglow Room-Temperature Emission from an Isolated Single Aromatic Unit under Ambient Condition. *Adv. Opt. Mater.* **5**, 1600996 (2017).

6. Authors show a schematic diagram to explain processes after excitation for Tb-dpph in Figure 4b. Readers cannot well image each rate of processes after the T_1 generation of dpph ligand. Readers would like to see the deactivation rate from T_1 in dpph ligand, the rate of endothermic energy transfer from T_1 of dpph ligand to 5D_4 of lanthanide center, and the rate of emission from 5D_4 of lanthanide center for Ln-dpph crystal as well as Tb-dpph crystals. The similar kinetics including endothermic process have been well evaluated in the field of thermally activated delayed fluorescence, such characteristics helps readers to well and accurately understand the kinetics of excited state of two crystals based on the concrete values.

<Answer 6> To address the reviewer's comment, information on the T_1 lifetime has been added to the energy diagram. An estimation of the exact internal emission lifetime of $^5D_4 \rightarrow ^7F_J$ was difficult because of the formation of excited-state equilibrium between T_1 and 5D_4 . Based on Tb emission lifetime data, the Tb(III) emission lifetime was expressed as " < 1.0 ms" to aid the understanding of the excited-state dynamics. In addition, the excited-state equilibrium suggested a larger energy-transfer rate constant than the inverse of the T_1 and 5D_4 lifetimes. Therefore, we have modified the energy diagram and added an explanation in the figure caption as follows:

<Figure 4 in manuscript>

Figure 4. Emission (solid line) and excitation (broken line) spectra of **Tb-dpph** (a: $\lambda_{\text{ex}} = 356 \text{ nm}$; $\lambda_{\text{em}} = 548 \text{ nm}$; 293 K). Excited-state dynamics of **Tb-dpph** (b). The photosensitized emission pathway via S_1 (black solid arrow) and T_1 (red solid arrow) states. The formation of an excited-state equilibration indicates that the rate constant for the energy transfer between T_1 and 5D_4 has been assumed to be much greater than 1 ms^{-1} .

7. How about emission yield of Ln-dpph crystal? How about emission spectral intensity at 1 ms, 10 ms, 100 ms, and 1s after stopping excitation without normalizing the spectral intensity for Ln-dpph crystal? This is important for readers to image performance of persistent emission. Current information in Figure 3c is not kind for audience to see information of decay time less than 100 ms.

<Answer 7> Thank you for your good suggestion. To address the reviewer's comment, we performed the experiments and added the results to the Supporting Information as follows. In addition, the emission images were re-estimated using high-resolution camera (PENTAX, K-70) carefully.

<Manuscript, P11>

The emission photograph of **Lu-dpph** is shown in Figure 3b, where it shows a green persistent luminescence.

Figure 3. The emission spectrum of **Lu-dpph** (a: $\lambda_{\text{ex}} = 400$ nm; delay: 80 ms; 100 K). Emission images of **Lu-dpph** excited using UV-light (b: $\lambda_{\text{ex}} = 375$ nm; 293 K) under vacuum condition. Emission decay curves of **Lu-dpph** (c: $\lambda_{\text{ex}} = 400$ nm; $\lambda_{\text{em}} = 530$ nm; 293 K).

<Supporting Information, P3>

Supplementary Note 1. Emission properties of Lu-dpph as a crystal

The emission decay (log-log plot) is shown in Figure S5. The decay shape is similar to sum up the exponential decay and power-law decay, which might suggest the existence of intermediate states for phosphorescence.^{S1} The normalized emission spectra of **Lu-dpph** with/without delay time are shown in Figure S6. From the comparison of the two emission spectra (Figure S6, black line and red line), the emission spectra are expected to originate from the fluorescence band edge (Figure S4) and phosphorescence spectrum (Figure 3a) of the dpph moiety. The phosphorescence quantum yield of **Lu-dpph** under Ar condition ($\lambda_{\text{ex}} = 400$ nm) was estimated to be 1.3 %. The emission spectra for various delay times (10, 20, 30, 50, and 100 ms) are also shown in Figure S7. The emission spectral band (delay: 10 ms) was slightly red-shifted after increasing delay times (20, 30, 50, and 100 ms), which was attributed to the disappearance of the dpph fluorescence band edge.

Figure S5. The emission decay curves (b: $\lambda_{\text{ex}} = 400 \text{ nm}$; $\lambda_{\text{em}} = 530 \text{ nm}$) of **Lu-dpph** at 293 K.

Figure S6. The emission spectra of **Lu-dpph** excited by 400 nm at 293 K (delay time, black line: 0 ms, red line: 20 ms). Normalized via emission at 550 nm.

Figure S7. The emission spectra of **Lu-dpph** excited by 400 nm at 293 K (delay time, black line: 10 ms, red line: 20 ms, blue line: 30 ms, green line: 50 ms, and purple line: 100 ms) without normalization.

8. For example, how do emission characteristics of Ln-dpph in vacuum when it is doped with amorphous solid host with large T_1 energy such as ZEONEX? Persistent emission caused by a rapid emission after endothermic process have been reported in thermally activated delayed fluorescence with the S_1 - T_1 energy gap as well as long-lived T_1 state when they are often doped in solid host with large T_1 energy. If Ln-dpph molecule potentially has different points, they also will interest researchers in a variety of fields including organic opto-electronics fields.

<Answer 8> Thank you for your good suggestion. We have performed the experiments as per the reviewer's suggestions and included the results in the Supporting Information as follows:

<Supporting Information, P10>

Supplementary Note 4

Photophysical properties of Lu-dpph and Tb-dpph dispersed in β -estradiol at room temperature

We also investigated the photophysical properties of **Lu-dpph** and **Tb-dpph** doped in solid host. Herein, β -estradiol was selected as a rigid amorphous host with a relatively high T_1 levels ($22,580\text{ cm}^{-1}$).^{S13} **Lu-dpph** and **Tb-dpph** were doped in β -estradiol (20 and 3 wt.%, respectively) at 200 °C and the samples were cooled to room temperature. The emission spectrum and emission decay behavior of **Lu-dpph** in β -estradiol (Figure S12) was similar to that of **Lu-dpph** in the crystal state. The emission spectrum and emission lifetime ($\tau = 0.84\text{ ms}$) of **Tb-dpph** in β -estradiol (Figure S13) is also similar to those of **Tb-dpph** in crystal state. The excitation spectra exhibit a strong band at approximately 370 nm, indicating effective photosensitization from the dpph ligand. These results suggest that effective intermolecular interactions between Tb(III) complexes are not necessarily required for thermally assisted energy transfer from the lower T_1 level to the emitting level of Tb^{III} . Therefore, we consider that the combination of Tb^{III} and organic ligands with a low T_1 level and transition probability ($T_1 \rightarrow S_0$) in a rigid amorphous host would be one of the novel strategies for the preparation of bright luminescent Tb^{III} complexes.

Figure S12. The emission spectrum (a: $\lambda_{\text{ex}} = 370$ nm; delay: 20 ms) and decay curves (b: $\lambda_{\text{ex}} = 400$ nm; $\lambda_{\text{em}} = 530$ nm) of **Lu-dpph** in β -estradiol.

Figure S13. The emission (a: $\lambda_{\text{ex}} = 356$ nm, solid line) and excitation spectra ($\lambda_{\text{em}} = 548$ nm, broken line) of **Lu-dpph** in β -estradiol. Inset: Emission decay curve (b: $\lambda_{\text{ex}} = 356$ nm; $\lambda_{\text{em}} = 548$ nm).

S13. Hirata, S. et al. Efficient persistent room temperature phosphorescence in organic amorphous materials under ambient conditions. *Adv. Funct. Mater.* **23**, 3386 (2013).

9. In Figure S6, why emission intensity decreases at temperature more than 300K?

<Answer 9> Thank you for your comments. We believe that the decrease in the emission intensity is related to the temperature-dependent T_1 lifetime in the dpnh moiety. We have added the explanation in the Supporting Information as follows:

<Supporting Information, P5>

A decrease in the emission intensity with increasing temperature (300 – 400 K) is considered to be originated from the increased non-radiative rate constant of $T_1 \rightarrow S_0$ transition in dpnh ligand.

10. Authors see thermally activated signal of emission in Figure S6. How about the estimated activation energy of the endothermic energy transfer? The activation energy appropriately explains the energy difference (650 cm^{-1})?

<Answer 10> Thank you for your good suggestion. As suggested by the reviewer, we tried to incorporate the activation-energy analysis using previously reported equations. However, multistep/several photosensitization pathways, such as intersystem crossing and energy transfer from the S_1/T_1 states, made it difficult to determine the activation energy. We will attempt to analyze this in the future. We have modified the text related to the temperature-dependent emission intensity as follows:

<Manuscript, P13>

The emission intensity increased with the temperature from 100 to 300 K, indicating the existence of a thermally enhanced photosensitization pathway such as intersystem crossing³⁸ and/or energy transfer from T_1 .

11. Because readers working in other fields are not familiar with the emission yield of other lanthanide complexes, the sound of the emission yield of 73% in Tb-dpnh crystal should be compared with previously reported other materials using lanthanide complexes.

<Answer 11> Thank you for your good suggestion. To address the reviewer's concern, we have added the explanation in the supporting information as follows:

<Supporting Information, P12>

Supplementary Note 5

Advantages of photosensitization with a low T_1 level for enhancement of brightness

Tb(III) complexes with high emission quantum yields ($\geq 70\%$) have been already reported.^{S14-S19} The focus of this study was achieving highly efficient photosensitization using a donating ligand with a low T_1 level, which resulted in the excitation of the low S_1 level, resulting in an effective low-energy-driven photosensitization.^{S20} The energy-accepting level of the Tb^{III} ion was relatively high (5D_4 : 20,500 cm^{-1}), requiring significantly high T_1 and S_1 level in the organic ligands for effective photosensitization. Brightness is defined by the product of molar absorption coefficient and emission quantum yield.^{S21} For example, [Tb(tmh)₃(tppo)] also produces a highly emission quantum yield ($\Phi = 66\%$)^{S3} but its brightness, when excited using UV-light ($\lambda = 365$ nm), is estimated to be approximately 6 $\text{M}^{-1} \text{cm}^{-1}$, based on the molar absorption coefficient (Figure S14a, $\epsilon_{365\text{nm}} = 9 \text{ M}^{-1} \text{cm}^{-1}$). In contrast, the brightness of **Tb-dpph** excited using UV-light ($\lambda = 365$ nm) was estimated to be 1,300 $\text{M}^{-1} \text{cm}^{-1}$ (Figure S14b, $\epsilon_{365\text{nm}} = 1,780 \text{ M}^{-1} \text{cm}^{-1}$). Thus, our concept is expected to be useful for improving light-absorption properties and brightness.

Figure S14. The electronic absorption spectra of [Tb(tmh)₃(tppo)] in toluene (a, $5.0 \times 10^{-3} \text{ M}$) and **Tb-dpph** in chloroform (b, $5.6 \times 10^{-4} \text{ M}$).

Reviewers' comments:

Reviewer #1 (Remarks to the Author):

Kitagawa and collaborators have adequately addressed my questions and substantially improved the work.

I recommend the acceptance of the manuscript.

Reviewer #2 (Remarks to the Author):

I appreciate authors for improving their manuscript and addressing some points that I have questions. However, I still have some unclear points regarding the current conclusion from authors. I would like to ask the following points if this manuscript shows critical data that authors explain.

(For the response to questions 3 and 7)

Authors introduced emission spectral intensity change depending on delay time for Lu-dpph in Figure 3a and Figures S5-S7. When phosphorescence spectrum of organic molecules does not have much vibrational structures, T1 energy of the molecules are determined from the onset energy of the phosphorescence spectra based on Frank-Condon theory (For example, supporting materials in J. Mater. Chem. C 2014, 2, 421) to discuss the relationship between the activation energy and the energy difference. Because considerable broad spectra are observed in Lu-dpph-phosphorescence, approximated onset energy of the broad Lu-dpph-phosphorescence (approximately 475 nm-480 nm) is slightly large compared with the excited state energy of Tb center. Therefore, I am not clear if the current estimation is accepted by broad audience to generally determine T1 energy of molecules. In Lines 10-12 in page 10, Authors explained $n-n^*$ transition for the phosphorescence. However, the spectral width is quite large and no significant vibrational structure of the phosphorescence spectra. Therefore, I am afraid if readers consider the $n-n^*$ transition for the phosphorescence. In revised manuscript, authors added Figure S10 to explain the temperature dependent emission lifetime characteristics. In kinetics of the endothermic process from a lower energy state with small radiation rate to a high energy state with large radiation rate (For instance, Figure 11a and 16 in FB Dias, TJ Penfold, AP Monkman, Methods and applications in fluorescence 5 (1), 012001), considerably long radiative decay characteristics due to long lived-lower energy state is often observed in low temperature. With elevated temperature, the decay intensity of short lifetime-region increases while the decay intensity of long lifetime-region decreases (Readers also would like to see temperature dependent emission decay characteristics). However, the black plots in Figure S10 of current revised manuscript do not likely show this kind of temperature dependent tendency. Therefore, I think future broad audience wonders the point and expects a distinct data showing the large contribution of T1 of ligand to Tb center to enhancement of emission yield with elevated temperature.

(For the response to question 10)

From response to question 7 from reviewer 1, I understand that the intersystem crossing (ISC) of S1 to triplet states of ligand as well as the energy transfer from T1 of the ligand to Tb center are potential endothermic processes. However, without a quantification of the magnitude regarding the contribution of the ISC from S1 in the ligand, I feel it is still difficult to reach the current conclusion that much contribution of the thermally-assisted process from T1 of ligand to Tb center to the enhanced emission. Therefore, discussions using identified activation energy are logically necessary to conclude the current conclusion that Authors insist. Otherwise, temperature dependent analysis regarding the ISC from S1 might be necessary.

Other minor comments for the points that authors added in their revised manuscript, I could not see explanation about the energy level of Ln center. This is also kind for readers to well understand this content.

Responses to the referee's comments and changes made in the revised manuscript

First, we would like to thank the referees and the editor for their valuable comments, based on which, we have carefully revised our paper. We have addressed all the comments and the detailed point-by-point responses to the comments are provided below.

Reviewer #1 (Remarks to the Author):

1. Kitagawa and collaborators have adequately addressed my questions and substantially improved the work. I recommend the acceptance of the manuscript.

Response: We appreciate the positive comments that helped us improve the quality of the manuscript.

Reviewer #2 (Remarks to the Author):

I appreciate authors for improving their manuscript and addressing some points that I have questions. However, I still have some unclear points regarding the current conclusion from authors. I would like to ask the following points if this manuscript shows critical data that authors explain.

Response: We performed the additional experiments, calculations, and re-evaluated our manuscript accordingly.

(For the response to questions 3 and 7)

1. Authors introduced emission spectral intensity change depending on delay time for Lu-dpph in Figure 3a and Figures S5-S7. When phosphorescence spectrum of organic molecules does not have much vibrational structures, T_1 energy of the molecules are determined from the onset energy of the phosphorescence spectra based on Frank-Condon theory (For example, supporting materials in J. Mater. Chem. C 2014, 2, 421) to discuss the relationship between the activation energy and the energy difference. Because considerable broad spectra are observed in Lu-dpph-phosphorescence, approximated onset energy of the broad Lu-dpph-phosphorescence (approximately 475 nm-480 nm) is slightly large compared with the excited state energy of Tb center. Therefore, I am not clear if the current estimation is accepted by broad audience to generally determine T_1 energy of molecules.

In Lines 10-12 in page 10, Authors explained π - π^* transition for the phosphorescence. However, the

spectral width is quite large and no significant vibrational structure of the phosphorescence spectra. Therefore, I am afraid if readers consider the π - π^* transition for the phosphorescence.

Response: As you pointed out, we did not perform the detail assignment of electronic transition in the green phosphorescence. However, we performed the assignment using theoretical calculations, and the results indicate that the transition is according to the characteristic strong π - π^* transition. To further support the transition character from the lowest excited state to ground state, we measured phosphorescence spectrum of the free ligand. The estimated spectrum is almost same with that of **Lu-dpph**. We added the calculation data and phosphorescence spectrum of free ligand in the supporting information as follows.

<Supporting Information>

Supplementary Note 2. TD-DFT calculations for [Tb₂(tmh)₆(dpph)]

To analyze the S₀-T₁ transition characteristics, we performed TD-DFT calculation using the **Lu-dpph** structure obtained by single X-ray crystal analysis. The MWB60 basis set was adopted for Lu atoms^{S4-S5}, whereas the 6-31G(D) basis set was used for the other atoms. The S₀-T₁ transition is mainly composed of two electronic configurations (Figure S3), which is corresponding to the π - π^* transition. The T₁ level was 19,220 cm⁻¹, which is similar to the experimentally estimation value (19,850 cm⁻¹). On the other hand, the S₀-T₂ transition is assigned to charge transfer transition from the tmh to dpph ligand. The T₂ level was 22,370 cm⁻¹, indicating a relatively large energy gap between the T₁ and T₂ state ($\Delta E(T_1-T_2) = 3,150$ cm⁻¹). To further confirm the transition from the lowest excited state to ground state for **Lu-dpph**, the phosphorescence spectrum of the free dpph ligand was evaluated (Figure S4). The emission spectral shape and transition orbital characteristics evaluated by TD-DFT calculation (Figure S5) of the free dpph ligand are similar with those of **Lu-dpph**. Thus, the observed phosphorescence bands of **Lu-dpph** (Figure 3a) are mainly due to the localized π - π^* transition character of the dpph moiety.

Figure S3. Main electronic configurations of (a) S_0 - T_1 and (b) S_0 - T_2 transitions for **Lu-dpph**.

Figure S4. Phosphorescence spectrum of the dpph ligands, which were doped in β -estradiol (7 wt.%), at 293 K (delay time: 50 ms, $\lambda_{\text{ex}} = 370$ nm).

Figure S5 Main electronic configurations of S_0 - T_1 transitions for the dpph ligand. Un-stabilized dpph LUMO (-1.94 eV \rightarrow -1.52 eV) level by de-complexation, which are consistent with the blue-shifted phosphorescence band by de-complexation (Figure 3a and Figure S4).

2. In revised manuscript, authors added Figure S10 to explain the temperature dependent emission lifetime characteristics. In kinetics of the endothermic process from a lower energy state with small radiation rate to a high energy state with large radiation rate (For instance, Figure 11a and 16 in FB Dias, TJ Penfold, AP Monkman, *Methods and applications in fluorescence* 5 (1), 012001), considerably long radiative decay characteristics due to long lived-lower energy state is often observed in low temperature. With elevated temperature, the decay intensity of short lifetime-region increases while the decay intensity of long lifetime-region decreases (Readers also would like to see temperature dependent emission decay characteristics). However, the black plots in Figure S10 of current revised manuscript do not likely show this kind of temperature dependent tendency. Therefore, I think future broad audience wonders the point and expects a distinct data showing the large contribution of T_1 of ligand to Tb center to enhancement of emission yield with elevated temperature.

Response: As the reviewer suggested, the endothermic process (e.g., r-ISC or the excited state equilibrium with long-lived lower excited state) generally induces the long-lived decay characteristics. In this case, the present energy transfer is an unusual thermally assisted energy transfer, which corresponds to the “exothermic” process (not “endothermic energy transfer”) as reviewer 1 pointed out [one of the models is based on the population of the 7F_5 level (Ref. *J. Lumin.*, 248, 118933 (2022), the other possible model is also discussed in the SI]. In the Supporting Information, we have added more detailed explanation from the excited state equilibrium perspective using the equation. Further, we have added decay data in the

supporting information.

<Supporting Information>

Supplementary Note 5

Temperature-dependent emission lifetimes for Tb-dpph and their comparison

We measured the temperature-dependent emission lifetimes using **Tb-dpph** (Figure S16-S17). The other data (Figure S18, [Tb(tmh)₃(tppo)]^{S9} (tppo: triphenylphosphine oxide) and [Tb₂(tmh)₆(bpeb)] (bpeb: 1,4-bis(diphenylphosphoryl)ethynylbenzene)) are also shown in Figure S16 for comparison. The T₁ level of the extended π -conjugated bpeb ligand was 21,230 cm⁻¹ from the TD-DFT calculations (B3LYP/6-31G(D)) using the structure optimized by DFT calculations (B3LYP/6-31G(D)). The emission lifetimes of [Tb(tmh)₃(tppo)] and [Tb₂(tmh)₆(bpeb)], with the extended π -conjugated systems, are insensitive and sensitive to temperature, respectively. The temperature-dependent emission lifetime of [Tb₂(tmh)₆(bpeb)] can be attributed to the deactivation pathway induced by back energy transfer from ⁵D₄ to T₁.^{S10} In contrast, **Tb-dpph** has a large π -conjugated phenanthrene framework with a low T₁ level (19,850 cm⁻¹), and its emission lifetime is insensitive to temperature in the thermally-enhanced emission region (100 K – 350 K). In the excited dynamics system, an excited-state equilibrium between the T₁ and ⁵D₄ levels was revealed by the oxygen-dependent emission lifetime. These results indicate the existence of a mechanism to suppress the quenching effect via the T₁ state with an excited-state equilibrium.

The equation of emission lifetime (τ_{obs}) in the excited equilibrium between donor and acceptor is generally expressed as follows.^{S11-S12}

$$\frac{1}{\tau_{obs}} = \alpha \frac{1}{\tau_{donor}} + (1 - \alpha) \frac{1}{\tau_{acceptor}} \quad (\text{eq. S1})$$

$$K_{eq} = \frac{\alpha}{1 - \alpha} = \frac{k_{A \rightarrow D}}{k_{D \rightarrow A}} \quad (\text{eq. S2})$$

Herein, $k_{A \rightarrow D}$ and $k_{D \rightarrow A}$ are the energy transfer from acceptor (emission center) to donor and energy transfer from donor to acceptor, respectively. τ_{donor} and $\tau_{acceptor}$ correspond to the time constants for the decays of the excited donor and acceptor moieties, respectively. K_{eq} is the excited-state equilibrium constant. From the equation, an excited-state equilibrium, with an energy-donating state having a lower energy level than the energy-accepting state and a significantly lower deactivation rate to the ground state than that from the accepting state to the ground state, should provide a significantly longer lifetime; however, the emission lifetime of **Tb-dpph** is only slightly longer than that of other Tb(III) complexes containing tmh and phosphine oxide ligands (Figure S16). From the eq. S1, a larger value of $k_{D \rightarrow A}$ than

that of $k_{A \rightarrow D}$ is a required condition for the temperature-insensitive emission lifetime in the excited state equilibrium; therefore, the simple endothermic energy transfer model is not appropriate to explain the **Tb-dpph** excited state dynamics.

The results suggest unusually efficient exothermic energy transfer pathways corresponding to the ${}^7F_5 \rightarrow {}^5D_4$ transitions from the T_1 states ($+\Delta E = 1,400 \text{ cm}^{-1}$) besides the endothermic energy transfer pathways corresponding to the ${}^7F_6 \rightarrow {}^5D_4$ transitions from the T_1 states ($-\Delta E = 650 \text{ cm}^{-1}$) (Figure 4b; the energy levels have been summarized in Table S2). The 7F_5 level is significantly populated owing to a long decay lifetime of ${}^7F_5 \rightarrow {}^7F_6$ in a relatively large energy gap between them ($2,050 \text{ cm}^{-1}$),^{S13-S16} allowing energy transfer from the 7F_5 level.^{S17} Theoretical calculations also suggest a larger matrix element for the energy-transfer rate of the ${}^7F_5 \rightarrow {}^5D_4$ transition than that of the ${}^7F_6 \rightarrow {}^5D_4$ transition.^{S18} Thus, the exothermic energy transfer corresponding to the ${}^7F_5 \rightarrow {}^5D_4$ transition from the T_1 states is potential pathways for the construction of the temperature-insensitive 4f-4f emission lifetime property. The other possibility is the effective energy transfer pathway from the T_2 state induced by thermally activated reverse internal conversion^{S19} from T_1 . Theoretical calculation suggests that the T_2 level ($22,370 \text{ cm}^{-1}$) is higher than the emitting level (5D_4 ; $20,500 \text{ cm}^{-1}$) from 7F_6 . Considering the experimental and theoretical aspects, the characteristic thermally assisted photosensitized emission occurs *via* the dpph T_1 state. This is the first example of the efficient photosensitized emission *via* the T_1 state in the lanthanide complexes with a T_1 level of an organic ligand lower than the emitting level of a Ln(III) ion.

Figure S16. Temperature-dependent emission lifetimes ($\lambda_{\text{ex}} = 355 \text{ nm}$, $\lambda_{\text{em}} = 540 \text{ nm}$) for **Tb-dpph** (black dot), $[\text{Tb}(\text{tmh})_3(\text{tppo})]$ (red dot)^{S9}, and $[\text{Tb}_2(\text{tmh})_6(\text{bpeb})]$ (blue dot).

Figure S17. Temperature-dependent emission decays for **Tb-dpph** ($\lambda_{\text{ex}} = 355\text{nm}$, $\lambda_{\text{em}} = 540\text{ nm}$) (a, black line: 100 K, red line: 150 K, blue line: 200K) (b, black line: 250 K, red line: 300 K, blue line: 350 K, pink line: 400K).

Figure S18. Chemical structures of (a) [Tb(tmh)₃(tppo)] and (b) [Tb₂(tmh)₆(bpep)].

Table S2. Energy levels of ligands and Tb(III).

	Energy / cm ⁻¹
S ₁ energy (dpph)	27,100
T ₁ energy (dpph)	19,850
S ₁ energy (tmh)	30,400
T ₁ energy (tmh)	24,400
⁵ G ₆ (Tb)	26,510 (24,460) ^a
⁵ D ₃ (Tb)	26,320 (24,270) ^b
⁵ D ₄ (Tb)	20,500 (18,450) ^c

a: Energy gap between ⁵G₆ and ⁷F₅. b: Energy gap between ⁵D₃ and ⁷F₅. c: Energy gap between ⁵D₄ and ⁷F₅.

Synthesis of [Tb₂(tmh)₆(bpep)] (Figure S18b)

[Tb₂(tmh)₆] (0.87 g, 0.6 mmol), and phosphine oxide ligand (bpep^{S20}: 0.32 g, 0.6 mmol) were dissolved in methanol. The solution was refluxed for 12 h. The solution was concentrated by an evaporator, and then filtrated with addition of small amounts of methanol into the solution. That mixture was left at room temperature to recrystallize giving colorless crystals (Yield: 69%, 0.83 g, 0.41 mmol).

Elemental analysis calcd.(%) for C₁₀₀H₁₃₈O₁₄P₂Tb₂: C, 61.79; H, 7.16. Found C, 61.70; H, 7.16.

(For the response to question 10)

3. From response to question 7 from reviewer 1, I understand that the intersystem crossing (ISC) of S_1 to triplet states of ligand as well as the energy transfer from T_1 of the ligand to Tb center are potential endothermic processes. However, without a quantification of the magnitude regarding the contribution of the ISC from S_1 in the ligand, I feel it is still difficult to reach the current conclusion that much contribution of the thermally-assisted process from T_1 of ligand to Tb center to the enhanced emission. Therefore, discussions using identified activation energy are logically necessary to conclude the current conclusion that Authors insist. Otherwise, temperature dependent analysis regarding the ISC from S_1 might be necessary.

Response: According to your suggestion, we re-considered the appropriate estimation method of thermally assisted emission properties without intersystem crossing contribution. We measured temperature-dependent emission intensity by “selective excitation of 4f-4f transition” for the direct demonstration of the thermally assisted energy transfer from the dpph triplet state to 5D_4 (Tb) and re-measured temperature-dependent emission intensity by excitation of π - π^* transition (dpph) using the rigid fixed cryostat system. The photophysical data showed the thermally enhanced emission intensity based on the excited state equilibrium between dpph triplet state and 5D_4 . The excited state equilibrium between the dpph triplet and 5D_4 (Tb) suggests the large contribution of the thermally enhanced energy transfer pathway from the triplet state, which is independent of the existence of the energy transfer pathway from dpph S_1 to Tb. From the whole results, we made a changed word from “efficient” to “effective” in one of the sentences in conclusion part.

Supplementary Note 4

Temperature-dependent emission intensity for Tb-dpph

Temperature-dependent emission spectra (at 100, 150, 200, 250, 300, 350, and 400 K) excited by the dpph ligand ($\lambda_{\text{ex}} = 356$ nm) were evaluated for **Tb-dpph** (Figure S12). The calculated emission area increased with an increase in temperature (Figure S13). The increased emission intensity with an increase in temperature (100–350 K) indicates the existence of thermally enhanced photosensitized emission pathways such as intersystem crossing^{S7} and/or energy transfer from T_1 . Non-increase in the emission intensity with increasing temperature (350–400 K) is considered to be originated from the increased non-radiative rate constant of $T_1 \rightarrow S_0$ transition in the dpph ligand.

Figure S12. Temperature-dependent emission spectra ($\lambda_{\text{ex}} = 356 \text{ nm}$, a: 100 K, b: 150 K, c: 200 K, d: 250 K, e: 300 K, f: 350 K, g: 400 K).

Figure S13. Temperature-dependent emission area ($\lambda_{\text{ex}} = 356 \text{ nm}$). Normalized by maximum value at 350 K.

To clarify the excited state dynamics related to T_1 and 5D_4 levels, the emission properties by direct 4f-4f excitation of the Tb(III) ion ($\lambda_{\text{ex}} = 482$ nm) without the intersystem crossing pathway ($S_1 \rightarrow T_1$) were evaluated for **Tb-dpph**. The temperature-dependent emission spectra (Figure S14; at 100, 150, 200, 250, 300, 350, and 400 K) were evaluated for **Tb-dpph** using direct 4f-4f excitation ($\lambda_{\text{ex}} = 482$ nm). The calculated emission area increased with an increase in temperature in the range of 100–350 K and decreased with a further increase in temperature from 350 to 400 K (Figure S15). These photophysical data show the thermally enhanced emission intensity in the excited state equilibrium between the dpph triplet and 5D_4 (Tb), which is revealed by oxygen-dependent emission lifetime (Figure S11).^{S8} Thus, the photo-sensitized emission pathway contains the thermally enhanced energy transfer pathway *via* the triplet state.

Figure S14. Temperature-dependent emission spectra ($\lambda_{\text{ex}} = 482 \text{ nm}$, a: 100 K, b: 150 K, c: 200 K, d: 250 K, e: 300 K, f: 350 K, g: 400 K).

Figure S15. Temperature-dependent emission area ($\lambda_{\text{ex}} = 482 \text{ nm}$). Normalized by maximum value at 350 K.

5. Conclusions

In this study, an effective photosensitized emission in a luminescent lanthanide complex with a T_1 level of an organic ligand lower than the emitting level of a Ln(III) ion was demonstrated for the first time.

Other minor comments for the points that authors added in their revised manuscript, I could not see explanation about the energy level of Ln center. This is also kind for readers to well understand this content.

Response: According to the reviewer's suggestion, we added the explanation about the energy level of Ln center in detail as follows.

Supplementary Note 1. Energy levels of Ln(III) center (Ln(III) = Tb(III) and Lu(III))

The electronic configuration of trivalent lanthanide ions is $[Xe]4f^n$ ($n = 0-14$). The $4f^n$ configurations generate various electronic levels characterized by three quantum numbers, S, L, and J. For Tb(III) ($[Xe]4f^8$), the J levels of the 7F and 5D terms are described by the Russell–Saunders coupling scheme. The possible J values for the 7F term are 0, 1, 2, 3, 4, 5, 6, so that the order of energies of the levels within the 7F term is $^7F_6 < ^7F_5 < \dots < ^7F_0$ based on the Hund's rule. Similarly, the lowest energy level of the 5D term is 5D_4 , which is corresponding to the emitting level of Tb(III). The Tb(III) emission bands are observed by the transition from the 5D_4 level to 7F_J ($J = 0, 1, 2, 3, 4, 5, 6$) level.^{S1-S2} The 4f-5d excited levels of Tb(III) ($>30,000 \text{ cm}^{-1}$) are higher than the dpph ligand S_1 and 4f-4f excited states^{S3}, which do not affect the excited state dynamics directly. The energy diagram of the Tb(III) ion is shown in Figure S1. In contrast, Lu(III) with a closed 4f-electronic configuration does not have the 4f-4f excited state. The 4f-5d excited energy levels of Lu(III) ($>80,000 \text{ cm}^{-1}$) is much higher than ligand excited state.^{S3}

Figure S1. Energy diagram of Tb(III).

Reviewers' comments:

Reviewer #2 (Remarks to the Author):

Authors almost addressed my questions. Concretely, authors added supporting data to well discuss the energy level of dpph and a hardly contribution of ISC process to the thermally increased emission characteristics. However, the temperature dependent-emission decay data of Tb-dpph still causes the following unclear points that general readers are difficult to follow.

As authors show, current temperature dependent-emission decay characteristics indicate exothermic characteristics of Tb-dpph. Although authors suggest that the transition from T1 of dpph Ligand to 7F5→5D4 is a potential pathway for the efficient exothermic energy transfer, the exothermic energy transfer process from ligand to metal may be generally common for this specific field. At least, readers will be still unclear why high emission quantum yield could be observed for Tb-dpph (Do authors suggest a considerable large exothermic rate compared with others for Tb-dpph?).

A hardly contribution of ISC process to the thermally assisted enhanced emission characteristics as well as a hardly change of emission decay characteristics depending on temperature have been explained with appropriate data evidence. However, now, readers cannot well understand a driving force to induce the approximate 2 times thermal enhancement of emission intensity from 100K to 350K in current content. Although a hardly temperature independent decay characteristics while a distinct enhancement of emission intensity with elevated temperature might be explained by a small increase of radiation rate while slight decrease of nonradiative rate with elevated temperature, science producing the temperature dependent characteristics of the radiation and non-radiation is unclear when we see the schematic diagram of Figure 4b.

From current data, thermally activated process from low T1 energy of dpph ligand to 7F6→5D4 energy of Tb center is not likely effectively utilized for the thermally assisted process. Therefore, the hardly utilization of low T1 energy of dpph ligand for thermally assisted emission enhancement and a somewhat unclear mechanism to allow the thermally assisted emission enhancement still preclude me to confidently recommend a publication in current stage.

Reviewer #3 (Remarks to the Author):

Although the authors have adequately addressed the questions raised by the two reviewers, the temperature dependence of the emission intensity and 5D4 lifetime of Tb-dpph, excited within the ligands and intra-4f, requires further clarification. As stressed by reviewer #2, the revised manuscript and SI do not provide evidence justifying these observed temperature dependencies (Fig. S12-S16) and, thus, this must be further addressed before recommending the publication of the paper.

The results described in the manuscript show that the T1->7F5-5D4 pathway dominates the ligand-to-Tb energy transfer in the complex. However, the role of the T1->7F5-5D4 pathway in the temperature dependency of the emission intensity although less important than the dominant pathway could not be discarded. The authors are encouraged to consider that possibility. In particular, the potential occurrence of a thermally activated phonon-assisted energy transfer mechanism involving the T1->7F6-5D4 that is dependent on the excitation energy (ligand or intra-4f level).

Responses to the referee's comments and changes made in the revised manuscript

First, we would like to thank the referees and the editor for their valuable comments, based on which, we have carefully revised our paper. We have addressed all the comments and the detailed point-by-point responses to the comments are provided below.

Reviewer #2 (Remarks to the Author):

Authors almost addressed my questions. Concretely, authors added supporting data to well discuss the energy level of dpph and a hardly contribution of ISC process to the thermally increased emission characteristics. However, the temperature dependent-emission decay data of Tb-dpph still causes the following unclear points that general readers are difficult to follow.

Response: We would like to thank the reviewer for taking the time to carefully review the manuscript. We have compiled point-by-point responses to all the comments.

Q1. As authors show, current temperature dependent-emission decay characteristics indicate exothermic characteristics of Tb-dpph. Although authors suggest that the transition from T_1 of dpph Ligand to ${}^7F_5 \rightarrow {}^5D_4$ is a potential pathway for the efficient exothermic energy transfer, the exothermic energy transfer process from ligand to metal may be generally common for this specific field. At least, readers will be still unclear why high emission quantum yield could be observed for Tb-dpph (Do authors suggest a considerable large exothermic rate compared with others for Tb-dpph?).

Response 1: The emission mechanism of the lanthanide(III) complexes is characterized by photosensitized emission from organic ligand owing to very weak absorption in 4f-4f transitions. Emission quantum yield is classified into two categories: Photosensitized (ligand-excited) emission quantum yield and 4f-4f-excited emission quantum yield. The 4f-4f emission quantum yield, in the case of no-back energy transfer from lanthanide center to ligand, is characterized by radiative rate constant and non-radiative rate constant in the 4f-4f transition. Among the lanthanide complexes, Tb(III) complexes exhibit a large energy gap ($14,800 \text{ cm}^{-1}$) between the emitting level (5D_4) and highest accepting state in emission (7F_0), and hence, Tb(III) complexes have a low non-radiative rate constant and a relatively high 4f-4f emission quantum yield. In case of the Tb(III) complexes, the photosensitized energy transfer efficiency

is often a critical factor for high photosensitized emission quantum yield.

The next important point is energy transfer rate. Experimental and theoretical data of the Ln(III) complexes (especially in Eu(III) and Tb(III)) show that the energy transfer rate from T_1 to Ln(III) emitting state is often “much faster” than the deactivation rate of the T_1 state and emitting state in Ln(III) (For ex. *Chem. Phys. Lett.* 187, 263 (1991), *Adv. Theory Simul.* 4, 2000304 (2021)). In addition, 4f-4f forbidden transition provides long-lived 4f-4f excited state (millisecond order). This fact resulted in “Latva’s empirical law”, in the lanthanide photochemistry history (*J. Lumin.*, 75, 149-169 (1997), Citation number > 1600). They suggested a threshold for the energy level difference between energy of the T_1 level and the emitting level of Tb(III) ions (5D_4 : 20,500 cm^{-1}) for suppression of photon loss by back energy transfer (required energy gap between donor and acceptor in case of Tb(III) complexes > 1850 cm^{-1}). Tb(III) complexes with high emission quantum yields ($\geq 70\%$) have been already reported, but the ligand design have been restricted because of the required high T_1 level (22,350 cm^{-1}). On the other hand, oxygen-dependent 4f-4f emission lifetime measurements revealed the formation of excited state equilibrium between emitting level of Tb(III) ion and ligand T_1 state in some Tb(III) complexes. The fact indicate the T_1 lifetime control is also key factor for efficient photosensitized energy transfer.

From the photophysical viewpoints in the previous reports, the high photosensitized emission quantum yield for **Tb-dpph** can be primarily attributed to the following two reasons. (1) Tb(III) complexes show the fundamentally high 4f-4f emission quantum yield with a relatively slow vibrational quenching in the 4f-4f transition. (2) The long T_1 lifetime should allow the efficient use of Ln(III) emitting photons, even when the T_1 level is lower than 5D_4 , when an excited equilibration between T_1 and Ln(III) emitting states is formed. Although the explanation (2) was shown in introduction, explanation (1) is not presented enough in our paper. Thus, I added the basic explanation from the viewpoint of lanthanide photochemistry in the SI as follows.

<Supporting Information, P2>

Supplementary Note 2. Emission quantum yield for Tb(III) complexes

Trivalent lanthanide ions show weak absorption (molar absorption coefficient (ϵ) of 4f-4f transitions $< 10 \text{ M}^{-1} \text{ cm}^{-1}$). This limitation can be overcome by using organic compounds that exhibit high light-absorption ($\epsilon = 10^3$ to $10^5 \text{ M}^{-1} \text{ cm}^{-1}$) as ligands in Ln^{III} complexes. Photosensitized emission quantum yield can be expressed using the following equation:

$$\phi_{\text{tot}} = \eta_{\text{sens}} \times \phi_{\text{ff}} = \eta_{\text{sens}} \times \frac{k_{\text{r}}}{k_{\text{r}} + k_{\text{nr}}} \quad (1)$$

Here, Φ_{tot} and Φ_{ff} are the photosensitized (ligand-excited) and 4f-4f excited emission quantum yield, respectively, while η_{sens} , k_{r} , and k_{nr} are the efficiency of sensitization, radiative rate constant in 4f-4f transitions, and non-radiative rate constant in 4f-4f transitions, respectively. Among the lanthanide complexes, Tb(III) complexes show a relatively large energy gap ($14,800 \text{ cm}^{-1}$) between emitting level ($^5\text{D}_4$) and highest accepting state in emission ($^7\text{F}_0$), resulting in a low non-radiative rate constant and a highly 4f-4f excited emission quantum yield (Φ_{ff}) in the lanthanide complexes.^{S4} Thus, the photosensitized energy transfer efficiency is often a critical factor for high photosensitized emission quantum yield. The 4f-4f forbidden transition provides long-lived 4f-4f excited state (sub-millisecond~millisecond order). Latva gave the “empirical law” for strong Tb(III) emission with long-lived excited states, in the lanthanide photochemistry history.^{S5} They suggested a threshold for the energy level difference between energy of the T_1 level and the emitting level of Tb(III) ions ($^5\text{D}_4$: $20,500 \text{ cm}^{-1}$) for suppression of photon loss from back energy transfer (required energy gap between donor and acceptor in case of Tb(III) complexes $> 1850 \text{ cm}^{-1}$). On the other hand, oxygen-dependent 4f-4f emission lifetime measurements revealed the formation of excited state equilibrium between the emitting level of Tb(III) ion and ligand T_1 state in some Tb(III) complexes^{S6-S8}. Therefore, the rate of energy transfer between these two states was faster than the rate of deactivation from $^5\text{D}_4$ and T_1 toward the ground state. This shows that controlling the lifetime of T_1 phosphorescence is important for the efficient transfer of photosensitized energy.

S2. A hardly contribution of ISC process to the thermally assisted enhanced emission characteristics as well as a hardly change of emission decay characteristics depending on temperature have been explained with appropriate data evidence. However, now, readers cannot well understand a driving force to induce the approximate 2 times thermal enhancement of emission intensity from 100K to 350K in current content. Although a hardly temperature independent decay characteristics while a distinct enhancement of emission intensity with elevated temperature might be explained by a small increase of radiation rate while slight decrease of nonradiative rate with elevated temperature, science producing the temperature dependent characteristics of the radiation and non-radiation is unclear when we see the schematic diagram of Figure 4b. From current data, thermally activated process from low T1 energy of dpqh ligand to $^7F_6 \rightarrow ^5D_4$ energy of Tb center is not likely effectively utilized for the thermally assisted process. Therefore, the hardly utilization of low T¹ energy of dpqh ligand for thermally assisted emission enhancement and a somewhat unclear mechanism to allow the thermally assisted emission enhancement still preclude me to confidently recommend a publication in current stage.

Response 2: Thank you for your comment. The temperature-dependent emission intensity is linked to the temperature-dependent photosensitized efficiency. According to the comments of Reviewers 2 and 3, we did not provide a detailed explanation for thermally enhanced emission intensity. The discussion of the energy transfer model detailing the energy transfer process from (T₁, 7F_6) to (S₀, 5D_4) is important for explaining the temperature-dependent emission intensity behavior. The (T₁, 7F_6)→(S₀, 5D_4) process causes the deactivation of the photon use for emission because of the slow endothermic energy transfer process. The increased temperature improves the endothermic energy transfer rate, improving the deactivation ratio from (T₁, 7F_6) to (S₀, 7F_6). We added the simulation data of the temperature-dependent emission intensity from (T₁, 7F_6)→(S₀, 5D_4) for demonstrating their validity in the SI as follows.

<Supporting Information, P16>

Supplementary Note 7

In order to further support the enhancing effect of endothermic T₁-to- 5D_4 energy transfer on the emission intensity with increasing temperature, we fit the equation of the extent of enhancement on the quantum yield with increasing temperature $\Phi_{\text{tot}}(T) - \Phi_{\text{tot}}(100\text{K})$ to the actual data. In the given system in which back energy transfer occurs (diagram shown in Figure 4), the quantum yield can be expressed by the following equation.^{S24}

$$\Phi_{\text{tot}} = \frac{A_r\{\eta_{\text{ISC}}W_{\text{FET},T_1} + \eta_{\text{FET},S_1}(A_{T_1} + W_{\text{FET},T_1})\}}{(A_{T_1} + W_{\text{FET},T_1})(A_r + A_{\text{nr}} + W_{\text{BET}}) - W_{\text{BET}}W_{\text{FET},T_1}} \quad (\text{eq. S3})$$

A_r and A_{nr} are the total radiative and the internal conversion rate constants of the $^5\text{D}_4$ state, respectively. η_{ISC} and η_{FET,S_1} are the intersystem crossing and S_1 -to- $^5\text{D}_4$ energy transfer efficiency, respectively. A_{T_1} is the total relaxation rate constant of the T_1 state. W_{FET,T_1} and W_{BET} are the rate constants of T_1 -to- $^5\text{D}_4$ energy transfer and its reverse process, respectively. The W_{FET,T_1} can be expressed by the Marcus-like equation:

$$W_{\text{FET},T_1} = \frac{2\pi}{\hbar} |\langle S_0, ^5\text{D}_4 | V | T_1, ^7\text{F}_6 \rangle|^2 (4\pi\lambda k_B T)^{-\frac{1}{2}} \exp \left\{ -\frac{(\lambda + \Delta G_{T_1 \rightarrow ^5\text{D}_4})^2}{4\lambda k_B T} \right\} \quad (\text{eq. S4})$$

where, \hbar , k_B , and T is the Planck constant, Boltzmann constant, and temperature, respectively. $\langle T_1, ^7\text{F}_6 | V | S_0, ^5\text{D}_4 \rangle$ is the matrix element of the interaction energy, λ is the reorganization energy, and $\Delta G_{^5\text{D}_4 \rightarrow T_1}$ is the energy gap between the T_1 and $^5\text{D}_4$ states. T_1 -to- $^5\text{D}_4$ energy transfer is endothermic in **Tb-dpph**, which means that $\Delta G_{^5\text{D}_4 \rightarrow T_1}$ is a positive value of 650 cm^{-1} . Since the interaction energy between the two states $\langle S_0, ^5\text{D}_4 | V | T_1, ^7\text{F}_6 \rangle$ are the same regardless of the direction of the energy transfer process between the states, if we assume that the potential energy surfaces at the relevant energy scale are similar in shape (same λ), the reverse process W_{BET} can be expressed as:

$$W_{\text{BET}} = \frac{2\pi}{\hbar} |\langle S_0, ^5\text{D}_4 | V | T_1, ^7\text{F}_6 \rangle|^2 (4\pi\lambda k_B T)^{-\frac{1}{2}} \exp \left\{ -\frac{(\lambda - \Delta G_{^5\text{D}_4 \rightarrow T_1})^2}{4\lambda k_B T} \right\} \quad (\text{eq. S5})$$

It should be noted that the only difference between eq. S4 and eq. S5 is the sign on $\Delta G_{^5\text{D}_4 \rightarrow T_1}$ in the exponent.

Here, we make two assumptions:

- Total of intersystem crossing and direct energy transfer to the $^5\text{D}_4$ state from the S_1 state occurs at unity ($\eta_{\text{FET},S_1} + \eta_{\text{ISC}} = 1$). This may be a drastic assumption but can be reasonably true considering that the total quantum yield of **Tb-dpph** reaches over 80%, implying that the total energy transfer efficiency from the ligand to the Tb^{3+} ion is very efficient.
- The total relaxation rate constant of the $^5\text{D}_4$ state is the inverse of the emission lifetime at 100 K ($A_r + A_{\text{nr}} = 1130 \text{ s}^{-1}$).

With these assumptions, equation of $\Phi_{\text{tot}}(T) - \Phi_{\text{tot}}(100\text{K})$ is sufficiently simplified where T and $\Phi_{\text{tot}}(T) - \Phi_{\text{tot}}(100\text{K})$ are the variables, $A_r + A_{\text{nr}}$ and $\Delta G_{^5\text{D}_4 \rightarrow T_1}$ are fixed, allowing η_{ISC} , A_{T_1} , A_r (with $A_{\text{nr}} = 1130 - A_r$), $|\langle S_0, ^5\text{D}_4 | V | T_1, ^7\text{F}_6 \rangle|^2$, and λ to be the fitting parameters. Figure S19 shows

the experimental data of $\Phi_{\text{tot}}(T) - \Phi_{\text{tot}}(100\text{K})$ (calculated based on the quantum yield at 300 K and the emission intensity provided in Figure S12) along with the fitted line using the equation. This resulted in the value shown in Table S3. All of the values obtained are reasonable for lanthanide complexes. This result suggest that **Tb-dpph** has a higher quantum yield at higher temperatures due to thermally-activated energy transfer.

Figure S19. Experimental values of $\Phi_{\text{tot}}(T) - \Phi_{\text{tot}}(100\text{K})$ (dots), and their fitted line (red) with the equation based on Figure 4 model system.

Table S3. Fit results of the temperature dependence of quantum yield.

Parameters	Values
η_{ISC}	0.366
$A_{\text{T}1}$	60.2 s^{-1}
A_{r}	712 s^{-1}
$\langle S_0, {}^5\text{D}_4 V \text{T}_1, {}^7\text{F}_6 \rangle$	0.0108 cm^{-2}
λ	1238 cm^{-1}
Reduced χ^2	1.28×10^{-3}
R^2	1.000
Adjusted R^2	1.000

Reviewer #3 (Remarks to the Author):

Although the authors have adequately addressed the questions raised by the two reviewers, the temperature dependence of the emission intensity and 5D_4 lifetime of Tb-dpph, excited within the ligands and intra-4f, requires further clarification. As stressed by reviewer #2, the revised manuscript and SI do not provide evidence justifying these observed temperature dependencies (Fig. S12-S16) and, thus, this must be further addressed before recommending the publication of the paper.

The results described in the manuscript show that the $T_1 \rightarrow ^7F_5 - ^5D_4$ pathway dominates the ligand-to-Tb energy transfer in the complex. However, the role of the $T_1 \rightarrow ^7F_5 - ^5D_4$ pathway in the temperature dependency of the emission intensity although less important than the dominant pathway could not be discarded. The authors are encouraged to consider that possibility. In particular, the potential occurrence of a thermally activated phonon-assisted energy transfer mechanism involving the $T_1 \rightarrow ^7F_6 - ^5D_4$ that is dependent on the excitation energy (ligand or intra-4f level).

Response: We would like to thank the reviewer for taking the time to carefully review the manuscript. We fully agree with your comments. In the previous manuscript, we provided insufficient explanation for temperature-dependent emission intensity behavior. As mentioned by Reviewer 3, considering the contribution of the pathway from ($T_1, ^7F_6$) to ($S_0, ^5D_4$) explains the increased emission intensity due to increased temperature. We added the simulation data of the temperature-dependent emission intensity from ($T_1, ^7F_6$) to ($S_0, ^5D_4$) for demonstrating their validity in the SI (Supplementary Note 7).

Reviewers' comments:

Reviewer #2 (Remarks to the Author):

Authors explain thermally activated emission enhancement using an additional calculation model. In the model (equations S3-S5), the shape of $\Phi(T) - \Phi(100K)$ does not likely change depending on η_{isc} . $WFET, T1$ increases approximately from $6 \times 10^1 s^{-1}$ to $7 \times 10^4 s^{-1}$ with elevated temperature from 100K to 400 K. $WBET$ are approximately $7 \times 10^5 - 8 \times 10^5 s^{-1}$ and hardly dependent on temperature (Supporting Table showing temperature dependence of $WFET, T1 (s^{-1})$ and $WBET (s^{-1})$ will be kind for chemistry audience). Because $\Phi(T) - \Phi(100K)$ using the $WFET, T1(T)$ and $WBET(T)$ increases with temperature when $AT1$ is small, the equations indicate the direction of thermally activated emission enhancement explained by authors.

However, the following two points might be still inconsistent when authors explain the kinetics of Figure 4b to explain the thermally activated emission enhancement. First, I think authors use $\Phi(100K)$ and $\Phi(T) - \Phi(100K)$ as temperature-independent term and thermally activated term, respectively. Because η_{isc} hardly changes the shape of $\Phi(T) - \Phi(100K)$ when other factors in Table 3 are fixed, $\Phi(T) - \Phi(100K)$ increases based on a delayed components via the thermal equilibrium between the state 5D4 in Tb(III) and the state T1 in ligand (cyclic process between the state 5D4 and the state T1). Therefore, the use of this model may need that the increase of $\Phi(T) - \Phi(100K)$ could be experimentally observed as a delayed emission from Tb(III) via the thermal equilibrium between the state 5D4 in Tb(III) and the state T1 in ligand. However, no observed delayed component in some temperature ranges from 100K to 400K. Therefore, the inconsistency might still remain. Second, when the diagram in Figure 4b as well as the fitting parameters in Table 3 are viewed, $WBET (8 \times 10^5 s^{-1})$ is approximately 80 times larger than $Ar + Anr (1.1 \times 10^3 s^{-1})$. In this situation, readers are difficult to consider the origin of the much prompt temperature-independent decay component [$\Phi(100K)$: approximately 40% from data of Figure S13] because the back energy transfer between the 5D4 and the T1 mostly occur before the generation of the prompt emission from Tb(III).

Because authors explain the adequacy of the kinetics depicted in Figure 4b using the proposed model and it is one of main parts in this manuscript, appropriate response or revision are still necessary.

A minor point: $\ln(\text{intensity})$ of Figures in text and SI needs scale and numerical values.

Reviewer #3 (Remarks to the Author):

The authors appropriately answered the referees' concerns and I recommend the publication of the paper.

Responses to the referee's comments and changes made in the revised manuscript

We have addressed all the comments and the detailed point-by-point responses to the comments are provided below.

Reviewer #2 (Remarks to the Author):

Authors explain thermally activated emission enhancement using an additional calculation model. In the model (equations S3-S5), the shape of $\Phi(T) - \Phi(100K)$ does not likely change depending on η_{isc} . W_{FET, T_1} increases approximately from $6 \times 10^1 s^{-1}$ to $7 \times 10^4 s^{-1}$ with elevated temperature from 100K to 400 K. W_{BET} are approximately $7 \times 10^5 - 8 \times 10^5 s^{-1}$ and hardly dependent on temperature (Supporting Table showing temperature dependence of $W_{FET, T_1} (s^{-1})$ and $W_{BET} (s^{-1})$ will be kind for chemistry audience). Because $\Phi(T) - \Phi(100K)$ using the $W_{FET, T_1(T)}$ and $W_{BET(T)}$ increases with temperature when A_{T_1} is small, the equations indicate the direction of thermally activated emission enhancement explained by authors.

Response: We thank the reviewer for taking the time to carefully review the manuscript. I added the Table as follows.

Table S4. Fit results of the temperature dependence of quantum yield.

Parameters	100 K	150 K	200 K	250 K	300 K	350 K	400 K
W_{FET} / s^{-1}	3.9112×10^2	1.0089×10^4	4.9111×10^4	1.2377×10^5	2.2539×10^5	3.4173×10^5	4.6276×10^5
W_{BET} / s^{-1}	4.5066×10^6	5.1469×10^6	5.2717×10^6	5.2146×10^6	5.0907×10^6	4.9445×10^6	4.7945×10^6
$\Phi / \%$	45	51	54	63	73	83	84

<Question-1> However, the following two points might be still inconsistent when authors explain the kinetics of Figure 4b to explain the thermally activated emission enhancement. First, I think authors use $\Phi(100K)$ and $\Phi(T) - \Phi(100K)$ as temperature-independent term and thermally activated term, respectively. Because η_{isc} hardly changes the shape of $\Phi(T) - \Phi(100K)$ when other factors in Table 3 are fixed, $\Phi(T) - \Phi(100K)$ increases based on a delayed components *via* the thermal equilibrium between the state 5D_4 in Tb(III) and the state T_1 in ligand (cyclic process between the state 5D_4 and the state T_1). Therefore, the use of this model may need that the increase of $\Phi(T) - \Phi(100K)$ could be experimentally observed as a delayed emission from Tb(III) *via* the thermal equilibrium between the state 5D_4 in Tb(III) and the state T_1 in ligand.

However, no observed delayed component in some temperature ranges from 100K to 400K. Therefore, the inconsistency might still remain.

Response-1:

We appreciate your comments about delayed emission. The pertinent experimental result is regarding the “temperature-independent emission lifetime.” It is difficult to explain this phenomenon using the simplified excited-dynamics model based on the three excited states ((S₀, ⁵D₄), (S₁, ⁷F₆), and (T₁, ⁷F₆)). The excited-dynamics construction containing the other excited states, which allow thermal equilibration with their excited states and much faster energy transfer to (S₀, ⁵D₄), are required for explanation of the temperature-independent emission lifetime. One candidate state is (T₁, ⁷F₅) (as is suggested by Reviewer 1). The other candidate state is (T₂, ⁷F₆), which is predicted by energy level calculations using Gaussian software.

To explain all the experimental results quantitatively, a simulation model based on three or more excited states ((T₁, ⁷F₆), (T₁, ⁷F₅), (S₀, ⁵D₄), (T₂, ⁷F₆)) may be required, in addition to determining the formation process of (T₁, ⁷F₅). At the present stage, it is difficult to achieve the exact simulation model because of the complexity of the states. Therefore, the (T₁, ⁷F₅) and/or (T₂, ⁷F₆) states have not been described in Figure 4b. Although the complete dynamics were not determined, several arrows describing energy transfer are shown in Figure 4b. As pointed out by you, this description is confusing. Thus, the arrows in Figure 4b have been deleted and we modified several sentences as follows for clarity. In addition, the several keywords (in title, abstract, results and discussion, and conclusion) are changed from “thermally assisted-energy transfer” to “thermally assisted-photosensitized emission”.

<Figure 4b: The energy diagram for Tb-dpph.>

<Supporting Information, P12>

Thus, the exothermic energy transfer corresponding to the ${}^7F_5 \rightarrow {}^5D_4$ transition from the T_1 states is potential pathways for the construction of the temperature-insensitive 4f-4f emission lifetime property. The other possibility is the effective energy transfer pathway from the T_2 state induced by thermally activated reverse internal conversion^{S22} from T_1 . Theoretical calculations suggest that the T_2 level (22,370 cm^{-1}) in the ground state structure is close to the emitting level (5D_4 : 20,500 cm^{-1}) from 7F_6 . These data suggest the possibility of thermal equilibration between three and/or four electronic states ($T_1(-{}^7F_6)$, $T_1(-{}^7F_5)$ (or/and $T_2(-{}^7F_6)$), and 5D_4 (- S_0)) by thermal stress. Considering the experimental and theoretical aspects, the characteristic thermally-assisted photosensitized emission occurs *via* the dp^{ph} T_1 state.

<Supporting Information, P15>

In order to discuss the enhancing effect of endothermic T_1 -to- 5D_4 energy transfer on the emission intensity with increasing temperature, we fit the equation of the extent of enhancement on the quantum yield with increasing temperature $\Phi_{\text{tot}}(T) - \Phi_{\text{tot}}(100\text{K})$ to the actual data using the simplified model based on three excited energy levels (S_1 , T_1 and 5D_4).

<Question-2> Second, when the diagram in Figure 4b as well as the fitting parameters in Table 3 are viewed, W_{BET} ($8 \times 10^5 \text{ s}^{-1}$) is approximately 80 times larger than $A_r + A_{\text{nr}}$ ($1.1 \times 10^3 \text{ s}^{-1}$). In this situation, readers are difficult to consider the origin of the much prompt temperature-independent decay component [$\Phi(100\text{K})$: approximately 40% from data of Figure S13] because the back energy transfer between the 5D_4 and the T_1 mostly occur before the generation of the prompt emission from Tb(III). Because authors explain the adequacy of the kinetics depicted in Figure 4b using the proposed model and it is one of main parts in this manuscript, appropriate response or revision are still necessary.

Response-2: We thank you for your suggestion. In the case of the excited state dynamics exhibiting the highly efficient BET from 5D_4 to T_1 , the Tb emission from 5D_4 can be invoked because the deactivation rate from T_1 to S_0 is expected to be extremely slower at 100 K than that of the other transitions.

A minor point: Ln(intensity) of Figures in text and SI needs scale and numerical values.

Response-3: We thank you reviewer for your suggestions. According to these comments, all figures in the manuscript have been modified as such.

Reviewer #3 (Remarks to the Author):

The authors appropriately answered the referees' concerns and I recommend the publication of the paper.

Response: We would like to thank the reviewer for taking the time to accurately review the manuscript and for their positive feedback.